# Households' expenditures for solid waste management services: Influencing factors and deep insight

Ghulam Mustafa[1], Naveed Hayat[1], Bader Alhafi Alotaibi[2]*, Abou Traore[3]

1 Department of Economics, Division of Management and Administrative Science, University of Education, Lahore, Pakistan, 2 Department of Agricultural Extension and Rural Society, College of Food and Agriculture Sciences, King Saud University, Riyadh, Saudi Arabia, 3 Department of Community Sustainability, College of Agriculture and Natural Resources, Michigan State University, 328 Natural Resources Building, East Lansing, United States of America

* balhafi@ksu.edu.sa

## Abstract

This study aims to evaluate the existing solid waste collection and management system available to households in Pakistan and to explore the factors affecting households' cash payments for waste collection and disposal services. Robust least square regression is applied to household-level data from 16,155 households in the Pakistan Social Living Measurement Survey (PSLM) for 2019–2020. This method was chosen for its ability to handle outliers and provide more reliable estimates. On average, households pay PKR 214 (USD 1.01) per month for waste collection and disposal services. Households in Baluchistan and Khyber Pakhtunkhwa pay the highest amounts, while those in Sindh and Punjab pay less. Rural households pay more than urban households. Waste collection is primarily handled by private vans/carts, with doorstep collection being the most common method. The municipality's role in waste collection at the doorstep is limited. Public bins and waste collection points are accessible to 83 percent of households, but their distant locations and infrequent emptying pose significant problems. These limitations highlight the need for improved municipal involvement and infrastructure. Results indicate that household income, education of the household head, age of the household head, gender of the household head, number of earners in the household, doorstep waste collection via private van/cart, availability of bins or waste collection points, distance from waste disposal facilities, bin or waste collection point clearance duration, house ownership, dwelling type, and number of rooms significantly affect households' cash payments for waste collection services. To increase cash payments for waste collection services, waste management authorities should provide better and modern solid waste management systems. Upgrading existing systems can enhance households' willingness to pay for these services.

**Data availability statement:** All relevant data are within the manuscript and its Supporting information files.

**Funding:** This research was funded by Researchers Supporting (Project Number RSP2025R443), King Saud University, Riyadh, Saudi Arabia. These funds were received by author Bader Alhafi Alotaibi. The funder had no role in study design, data collection and analysis, decision to publish, or preparation of the manuscript.

**Competing interests:** The authors have declared that no competing interests exist.

## Introduction

The increasing volume and complexity of the waste associated with the modern economy poses a serious risk to ecosystems and human health [1]. Public access to reliable solid waste services is an essential ingredient for improved human health, safe environment and sustainable development [2]. Developed and developing countries face serious challenges in the solid waste management. However, the problem of solid waste management is more prominent in middle and low-income countries where there is rapid population growth and urbanization [3]. The booming growth of cities of the developing world has created limited financial resources of municipalities to deal with the provision of solid waste management services. This is because the rate of generation of solid waste increases with the increase in population, technological development and the changes of the life style of the people. Even though developed countries generate greater quantity of solid waste than developing countries, the problem of solid waste management in developing countries is more acute than developed countries [4].

Due to the rapid growth of economic, urbanization, industrialization and population in urban areas, the demand for solid waste management services have increased significantly [5]. According to the World Bank study, globally, the annual generation of solid waste exceeds 2 billion tonnes and this is expected to increase to 2.2 billion tonnes in 2025 [6–7]. According to the statistics of the United Nations, world population is expected to reach 9.7 billion in 2050; with this, the waste generation would be further exacerbated with further increase in population [8] with estimated solid waste of 3.40 billion tonnes [7]. Presently, only about 38 percent of solid waste is being managed in an environmentally friendly manner, while the rest is handled in an unsustainable way, resulting in environmental challenges such as ecosystem damage, abiotic resources depletion, human health hazards, ozone depletion, and global warming [8–9]. According to the World Bank study, cost of solid waste management is $205.5 billion which is expected to increase $375.5 billion in 2025 [6]. In developing countries, about 30 percent to 50 percent of municipal operation budget is spent on collection of waste and its management which covers less than 50 percent of the total population [4,10,11]. Similarly, households also spend a huge cash payment for solid waste management which is still undocumented yet. Therefore, this study is designed to find households cash payment for solid waste management.

The global average for waste generation is roughly 0.74 kg, with country-to-country variations ranging from 0.11 kg to 4.54 kg per person per day [12–13]. Usually, wastes are managed by incineration, landfills, dumps and composting processes. The increase in solid waste have many environmental repercussions such as greenhouse gas emission and devastating pollution. Unmanageable solid waste accumulation deteriorates human health, affects landscapes, spread diseases, and harms the urban environment [14–15]. For instance, it has been estimated that 3.3 billion tonnes of $CO_2$ is accumulating into the atmosphere every year due to food waste which is the major portion of municipal solid waste [16–17]. Even in case of recycling of solid waste, it is possible of potential health impacts through recuperation in circular economy supply chain [7].

There are number of factors that affect the households' cash payment to collect solid waste. Among others, income of household head is the major determinant of cash payment of solid waste management. The studies have empirically proved that financially well-off families has positive attitude toward waste management and hence their propensity to pay for waste disposal is higher [1,18]. Tadesse et al. [19] were of the opinion that families with higher incomes use communal waste containers and they are less likely to dispose of waste on unapproved places. Therefore, income can be considered strong predictor of cash payment of solid waste disposal.

Access to solid waste management services significantly impact the household cash payment to manage the waste. Endalew and Tassie [4] were of the opinion that access to waste management services create environmental awareness and their negative impacts on health among the respondents. Therefore, such services motivate the households to pay for waste collection services. Similarly, Chukwuone et al. [20] found that availability of dumpsters in the locality significantly reduce the likelihood of illegal waste disposal. If waste management services are not provided, it can create isolation among individual and create fear of liability [21]. Usually people are reluctant to pay for waste disposal and its management. However, they willing to spare their time for such services including cleanup activities, involvement in voluntary service and membership of an association. It can be measured in man-days or hours worked in cleaning or disposal activities. For instance, Chukwuone et al. [20] estimated that the number of man-days or hours they were willing to participate in a week on waste disposal activities. They found that man-days in cleaning activities significantly and positively impact the waste management. Most of times, households give time instead of cash payment: economists refer this as opportunity cost. In this case, their willingness to pay cash decreases due to involvement in cleaning or disposal of waste activities.

Efficient waste management is essential for both environmental and health of residents as inadequate management of solid waste poses the greatest risk to local municipality. However, due to budgeting and infrastructural constraints in developing countries most municipal authorities are often unable to manage large volumes of solid waste generated. Solid waste management system comprises waste generation, collection, transportation and disposal; thus, waste management requires adequate budgetary and infrastructural provision [22]. A significant portion of the municipal budget is spent on solid waste management in Asian countries but a rapid increase in population, economic growth, and improvement of living standard have resulted in the substantial increase in the amount of solid waste being generated, making solid waste management even more challenging [23]. Like other Asian countries, solid waste management is a huge problem for the national, provincial, and local governments of Pakistan as well.

In Pakistan, local and municipal governments are responsible for collecting waste throughout most of Pakistan's major cities. Solid waste collection by government owned and operated services in Pakistan's cities currently averages only 50 percent of waste quantities generated; however, for cities to be relatively clean, at least 75 percent of these quantities should be collected [24]. The garbage collection fleet typically is comprised of open trucks, tractor/trolley systems, and arm roll containers/trucks for secondary collection and transfer. Handcarts and donkey pull-carts are used for primary collection. Some municipalities hire street sweepers and sanitary workers to augment other collection methods. They use wheelbarrows and brooms to collect solid waste from small heaps and dustbins, then store it in formal and informal depots [25].

Solid waste management capabilities and systems vary by province. In Punjab, Lahore is the only city with a proper solid waste management, treatment, and disposal system, which was outsourced to Turkish companies Albayrak and OzPak. In Sindh, Sindh Solid Waste Management Board (SSWMB) aims to improve solid waste management services in 20 cities. In Khyber Pakhtunkhwa, the Water and Sanitation Services Peshawar (WSSP) is planning to build a sanitary landfill. Balochistan, Pakistan's largest province by area but with a sparse population of 6.9 million, has no significant infrastructure for waste management. Much of Pakistan's solid waste is retrieved for recycling, primarily by scavengers, before it ever reaches disposal locations, and a large portion of the country's solid waste never makes it to final disposal sites [24–25].

However, the country's current system of municipal waste management is far from satisfactory. The services are, by and large, provided by municipalities and limited to partial collection and open dumping or burning [26]. Unfortunately, none of the cities in Pakistan has a proper solid waste management system right from collection of solid waste up to its proper disposal. Much of the uncollected waste poses serious risk to public health through clogging of drains, formation of stagnant ponds, and providing breeding ground for mosquitoes and flies with consequent risks of malaria and cholera. In addition, because of the lack of adequate disposal sites, much of the collected waste finds its way in dumping grounds, open pits, ponds, rivers and agricultural land [24,27].

Pakistan generates 30 million metric tons of municipal solid waste per year, which has been increasing at the rate of 2.4 percent annually. Moreover, a substantial increase in solid waste in the coming years is expected due to rapid population growth, urbanization, and economic development in the country. However, like other developing countries, Pakistan lacks proper waste management infrastructure which create serious environmental problems in the country. Most municipal waste is either burned, dumped, or buried on vacant lots, threatening the health and welfare of the common masses. The country is facing enormous challenges on how to manage solid waste. Bureaucratic hurdles, lack of urban planning, inadequate waste management equipment, and low public awareness contribute to the problem. Therefore, Pakistan urgently needs a waste road map for its policy makers, to make progress toward better health for its residents and reduce the contamination of land [26].

Due to the importance of the issue under consideration, this study aims to evaluate the existing solid waste collection and management system available to households in Pakistan. The study also explores the factors affecting the households' cash payments for waste collection and disposal services. Findings of the study will guide the policymakers in devising proper waste management policies at national, provincial and districts level to make the existing solid waste collection and management system more effective and to make progress toward better health for its people. After a detail introduction, the remainder of the paper is organized as follows: The Methodology section outlines the study site, data sources, analytical framework, and estimation strategy employed in this study. The Results and discussion section presents the study's findings, which are comparatively evaluated with the findings from similar studies conducted in Pakistan and other countries. Finally, the Conclusion and recommendations section summarizes the key findings and offers recommendations based on the analysis.

## Methodology

### Study site

Pakistan, officially the Islamic Republic of Pakistan, is a country in South Asia. It is the world's fifth most populous country, with a population of almost 241.49 million, and has the world's second largest Muslim population. In the country, 61 percent of the population live in rural areas whereas 39 percent of the population live in urban areas [28]. Pakistan is the 33rd largest country by area, spanning 881,913 square kilometers (340,509 square miles). It has a 1,046 kilometers (650-mile) coastline along the Arabian Sea and Gulf of Oman in the south. Pakistan is the federation of four provinces namely, Punjab, Sindh, Khyber Pakhtunkhwa, and Baluchistan as shown in the Fig 1.

Punjab has a population of about 127.68 million, according to the 2023 Population and Housing Census. In the province, the rural population is around 59 percent while the urban population is around 41 percent [28]. It has more people than the rest of Pakistan combined. According to the House-hold Integrated Economic Survey (HIES) 2018–2019, the average monthly income of households in Punjab is PKR 42,861 (1USD = PKR 220) and the average size of household is approximately 5.78 members [29]. Sindh is located in the southeastern region of the country. Sindh is the third-largest province of Pakistan by total area and the second-largest province by population. Sindh has a population of about 55.69 million, according to the 2023 Population and Housing Census. In the province, the rural population is around 46 percent while the urban population is around 54 percent [28]. According to the HIES 2018–2019, the average monthly income of

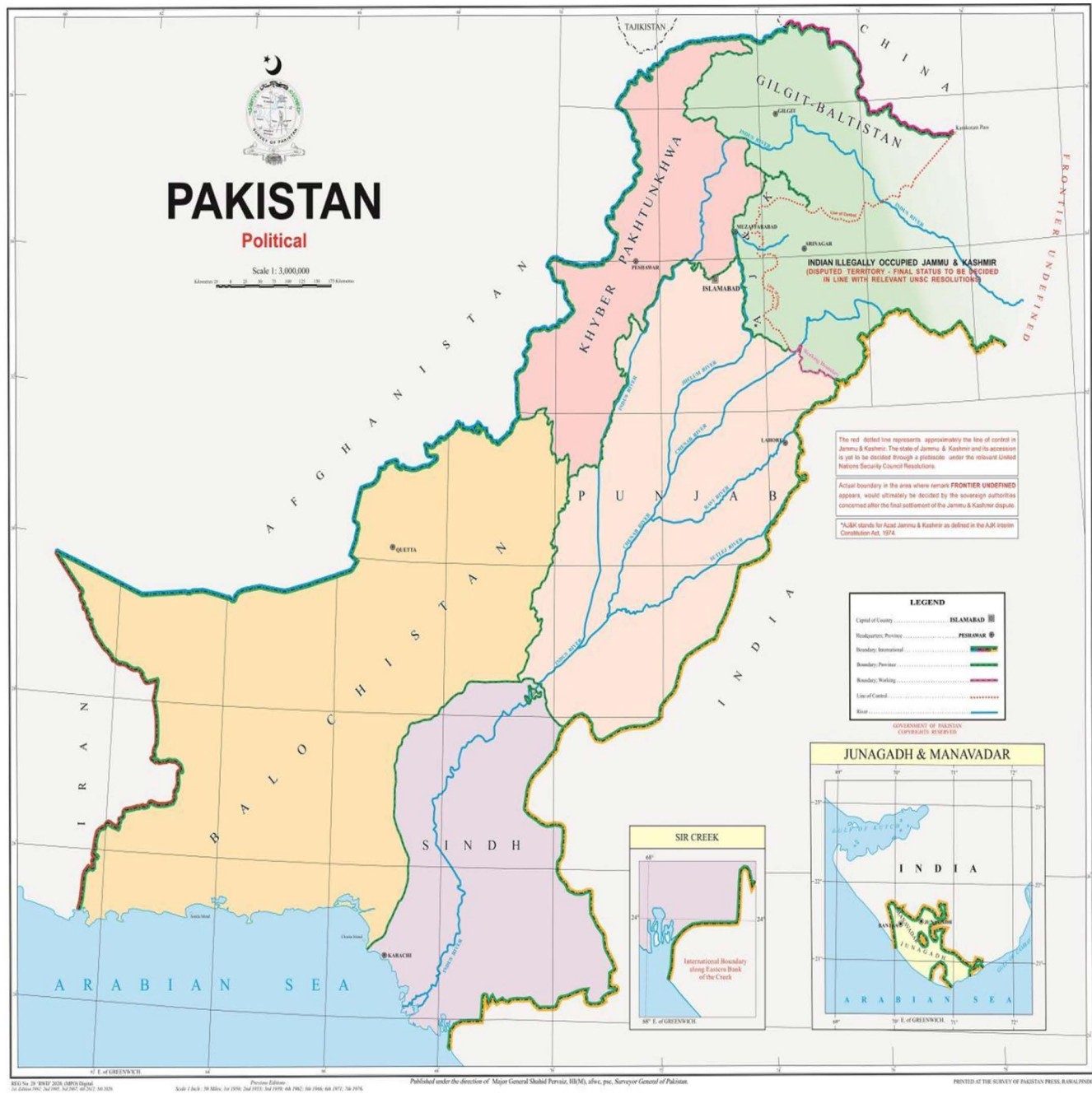

*Source:* Survey of Pakistan (2020).

**Fig 1. Map of the Study.** Survey of Pakistan (2020) provides the official *Political Map of Pakistan* (5th ed.). This map is published by the Ministry of Defence, Government of Pakistan, and is available at https://www.surveyofpakistan.gov.pk/index.

households in Sindh is PKR 39,078 (1USD = PKR 220) and the average size of household is approximately 6.23 members [29].

Khyber Pakhtunkhwa is located in the north western region of the country, along the Afghanistan-Pakistan border and close to Tajikistan border. Khyber Pakhtunkhwa is the third-largest Pakistani province in terms of both its population and its economy. Khyber Pakhtunkhwa has a population of about 40.85 million, according to the 2023 Population and Housing Census. In the province, the rural population is around 85 percent while the urban population is around 15 percent [28]. According to the HIES 2018–2019, the average monthly income of households in Khyber Pakhtunkhwa is PKR 42,736 (1USD = PKR 220) and the average size of household is approximately 7.41 members [29]. Baluchistan is the largest province in terms of land area, forming the southwestern region of the country, but it is the least populated province of the country. Baluchistan has a population of about 14.89 million, according to the 2023 Population and Housing Census. In the province, the rural population is around 69 percent while the urban population is around 31 percent [28]. According to the HIES 2018–2019, the average monthly income of households in Baluchistan is PKR 36,387 (1USD = PKR 220) and the average size of household is approximately 8.12 members [29]. To achieve our objective, we have selected Pakistan and its four provinces shown in Fig 2.

**Data**

This study used the recent data of the Pakistan Social Living Measurement Survey (PSLM) for the year 2019–2020, conducted by the Pakistan Bureau of Statistics. PSLM, 2019–20 is the twelveth round of a series of surveys, initiated in 2004. The first round of the survey started in 2004–2005 and upto date the most recent round of survey data was conducted in year 2019–20. The survey was not carried out for 2 rounds −2009–10 and for 2017–18 [29]. Besides in each round the sample size is changed. Thus, making a panel data from various round of PSLM is not possible. Thus, in the household level studies the researchers often prefer cross-sectional data over panel data [30]. Furthermore, we use the most recent data of the 12th round of PSLM for the year 2019–2020, which include 160,654 households throughout Pakistan. However, some households did not report the information on their cash payments for waste collection and disposal services, so after organizing the data, we used data from 16,155 households who pay cash for waste collection and disposal services.

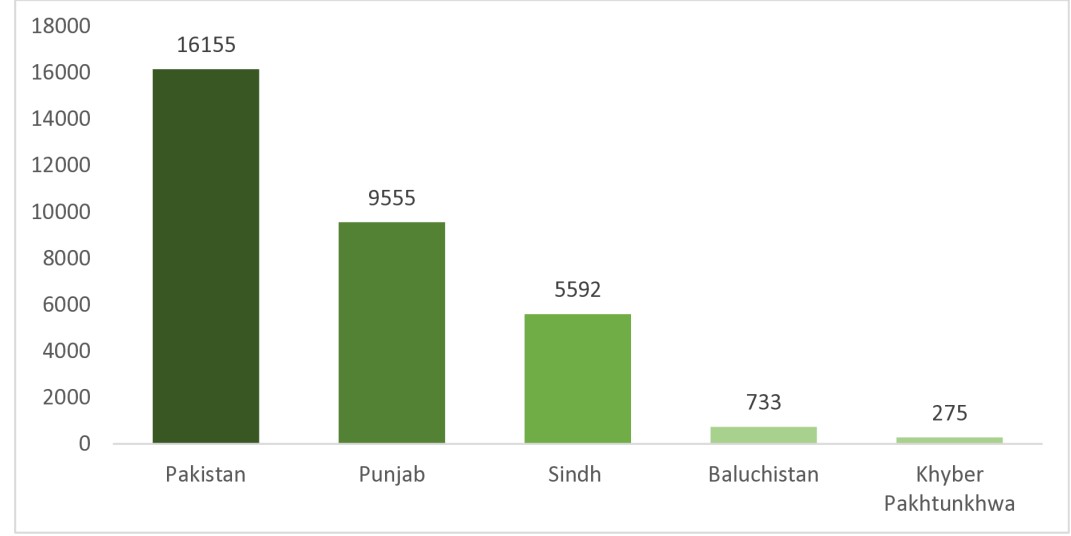

*Source*: Estimated by authors based on PSLM data for the year 2019-20.

**Fig 2. Province-wise number of households selected for the study.**

Fig 2 shows that out of 16155 households, 59.15% (9555) households belong to Punjab, 34.61% (5592) households belong to Sindh, 4.54% (733) households belong to Baluchistan, and only 1.70% (275) households belong to Khyber-Pakhtunkhwa. Similarly, out of 16155 households, 89.79% (14505) households living in urban areas whereas 10.21% (1650) household living in rural areas of the country as shown in Fig 3. The dataset provides information on the households' monthly cash payment for waste collection and disposal services, households monthly income, households waste management techniques, average time the households spent on a round trip to the nearest public bin/collection point, duration of the nearest public bin emptied/cleared, dwelling type of households, region, and province to which the households belong.

## Analytical framework and estimation strategy

**Analytical framework.** The analytical framework we employ for this study is based on a utility maximization problem proposed by Omotayo et al. [1] and Oyekale [18]. The only difference in our analytical framework of utility maximization problem and the utility maximization problem proposed by Omotayo et al. [1] and Oyekale [18] is that they included the recycling behavior into utility maximization problem while we did not included the recycling behavior. The utility maximization problem is used to derive the function for the empirical modelling of household decisions regarding the disposal of solid wastes through cash payments. Assume an economy of consisted of $N$ identical households with utility functions represented by:

$$u_i = u(x_i, \ e_i) \tag{1}$$

where $x_i$ is the composite goods and services consumed by $ith$ household and $e_i$ is the environmental quality that is being enjoyed by $ith$ household. Suppose the environmental quality function is expressed as:

$$e_i = e(g_i, \ h_i) \tag{2}$$

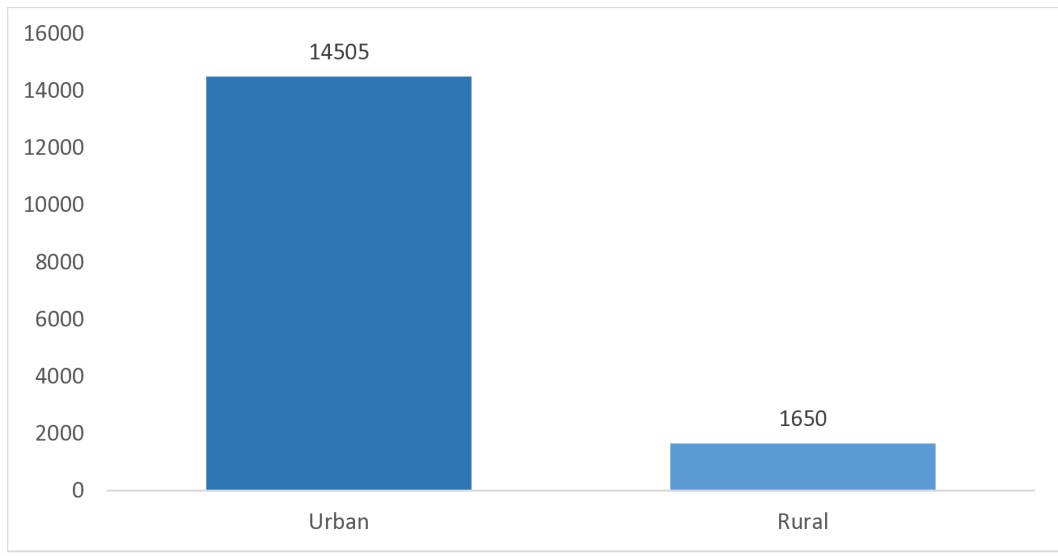

*Source*: Estimated by authors based on PSLM data for the year 2019-20.

**Fig 3. Region-wise number of households selected for the study.**

where $g_i$ is the quantity of solid wastes that was disposed by *ith* household by paying a specific cash payment on monthly basis for waste collection and disposal services. The values of $g_i$ in the environmental quality model $e_i$ is related to a set of some socio-demographic characteristics $h_i$ possessed by the households. If equation (2) is directly substituted into equation (1), the final expression of consumer's utility function is derived as:

$$u_i = u[x_i, \ e(g_i, \ h_i)] \tag{3}$$

This utility function given in equation (3) can be maximized subject to the budget constraint faced by the *ith* household. The expression of budget constraint is presented as:

$$x_i + p_g g_i = m_i \tag{4}$$

It should be noted that in equation (4), the price of $x_i$ is normalized to unity, $p_g$ is unit price of waste disposal, and $m_i$ represents total income of *ith* household. Solving the utility maximization problem, we get the following waste disposal function:

$$g_i = g(p_g, m_i, h_i) \tag{5}$$

**Empirical estimation strategy.** Although the dependent variable is binary (yes = 1 and 0 otherwise) in the empirical estimation strategy proposed by Omotayo et al. [1] and Oyekale [18]. However, in our model the dependent variable is continuous in nature (household monthly cash payment for waste collection and disposal services in Pakistani rupees). The nature of the dependent variable naturally suggests estimation of liner regression to determine the factors affecting waste disposal through cash payments. Therefore, we devise the following regression model based on equation (5):

$$g_i = \alpha + \beta m_i + \gamma_{ik} \sum_{i=1}^{k} h_i + \varepsilon_i \tag{6}$$

where $\alpha$, $\beta$, and $\gamma_i$ are the parameters to be estimated and $\varepsilon_i$ is the error term. In our model the dependent variable is household monthly cash payment for waste collection and disposal services [20], the explanatory variables include: household monthly income [1,4,20,31–33], household socio-demographic characteristics like education of household head [1,4,18,34,35], age of the household head [1,18,22,31,36–39], gender of household head [23,33,36,38,39], number of income earners in household, household waste management techniques [4,20,33], average time the household spent on a round trip to the nearest public bin/collection point [36,40], duration of the nearest public bin emptied/cleared [36,40], dwelling type [35,36], regional dummies, and provincial [35] dummies as explained in Table 1 (in next section). The studies carried out in the past like on the same issue under consideration like Omotayo et al. [1] and Oyekale [18] used primary data. Thus, they included various variables like education of household head, gender of household head, number of income earners in household in their empirical models. However, we used cross-sectional data collected by Pakistan Bureau of Statistics; thus, we used the available information provided by this data set. Therefore, instead of using the variables used by previous studies, we incorporated some interesting variables like household income, household waste management techniques, average time the household spent on a round trip to the nearest public bin/collection point, duration of the nearest public bin emptied/cleared, household housing characteristics, and dwelling type that are directly linked with the household monthly cash payment for waste collection and disposal services.

Since we use cross-sectional data which is often exposed to the problem of the heteroscedasticity or non-constant variance. The major cause of heteroscedasticity in the cross-sectional data is an outlying observation, or outlier, is an observation that is much different (either very small or very large) in relation to the observations in the sample. Thus, in the presence of heteroscedasticity due to presence of outlier in the data the estimation of regression model (6) via ordinary

**Table 1. Descriptive Statistics.**

| Variables | Definition | Mean | Standard Deviation | Minimum | Maximum |
|---|---|---|---|---|---|
| Dependent | | | | | |
| Household cash payment for waste collection and disposal services | Household monthly cash payment for waste collection and disposal services (PKR) | 214 | 277 | 50 | 5500 |
| Explanatory | | | | | |
| *Household income:* | | | | | |
| Income | Household monthly income (PKR) | 25497 | 53289 | 5000 | 5200005 |
| *Household socio-demographic characteristics:* | | | | | |
| Education of household head | Education of household head (Years) | 9 | 5.3 | 0 | 23 |
| Age of the household head | Age of the household head (Years) | 48 | 13.2 | 15 | 99 |
| Female household head | 1 if the household head is female, 0 otherwise | 0.08 | 0.27 | 0 | 1 |
| Number of income earners in household | Number of persons in the household who earn income | 1.4 | 0.76 | 1 | 8 |
| *Household waste management techniques:* | | | | | |
| Household waste been collected by private van/cart from door step | 1 if the household waste been collected by private van/cart from door step, 0 otherwise | 0.72 | 0.45 | 0 | 1 |
| Bin/waste collection point available/accessible | 1 if bin/waste collection point is available/accessible to household, 0 otherwise | 0.83 | 0.38 | 0 | 1 |
| *Average time the household spent on a round trip to the nearest public bin/collection point:* | | | | | |
| 1–5 minutes | 1 if the household spent 1–5 minutes on a round trip to the nearest public bin/collection point, 0 otherwise | 0.52 | 0.50 | 0 | 1 |
| 6–10 minutes | 1 if the household spent 6–10 minutes on a round trip to the nearest public bin/collection point, 0 otherwise | 0.19 | 0.39 | 0 | 1 |
| 11–15 minutes | 1 if the household spent 11–15 minutes on a round trip to the nearest public bin/collection point, 0 otherwise | 0.08 | 0.27 | 0 | 1 |
| 16–20 minutes | 1 if the household spent 16–20 minutes on a round trip to the nearest public bin/collection point, 0 otherwise | 0.03 | 0.16 | 0 | 1 |
| 21–25 minutes | 1 if the household spent 21–25 minutes on a round trip to the nearest public bin/collection point, 0 otherwise | 0.008 | 0.09 | 0 | 1 |
| *Duration of the nearest public bin/waste collection point emptied/cleared:* | | | | | |
| Every day | 1 if the nearest public bin/collection point is emptied/cleared every day, 0 otherwise | 0.42 | 0.49 | 0 | 1 |
| Once a week | 1 if the nearest public bin/collection point is emptied/cleared once a week, 0 otherwise | 0.11 | 0.32 | 0 | 1 |
| Twice a week | 1 if the nearest public bin/collection point is emptied/cleared twice a week, 0 otherwise | 0.06 | 0.24 | 0 | 1 |
| Thrice a week | 1 if the nearest public bin/collection point is emptied/cleared thrice a week, 0 otherwise | 0.3 | 0.17 | 0 | 1 |
| *Household housing characteristics:* | | | | | |
| Own house | 1 if the household living in his own house, 0 otherwise | 0.83 | 0.38 | 0 | 1 |
| On rent | 1 if the household living in a rented house, 0 otherwise | 0.10 | 0.31 | 0 | 1 |
| Number of rooms in the house | Number of rooms in the house | 2.3 | 1.3 | 1 | 12 |
| *Dwelling type:* | | | | | |
| Independent house | 1 if the household live in independent house, 0 otherwise | 0.77 | 0.42 | 0 | 1 |
| Apartment | 1 if the household live in apartment, 0 otherwise | 0.03 | 0.18 | 0 | 1 |
| Part of the large unit | 1 if the household live in part of a large unit, 0 otherwise | 0.13 | 0.34 | 0 | 1 |
| Part of compound | 1 if the household live in part of a compound, 0 otherwise | 0.07 | 0.25 | 0 | 1 |
| *Region:* | | | | | |
| Urban | 1 if the household belongs to urban area, 0 otherwise | 0.90 | 0.30 | 0 | 1 |

*(Continued)*

**Table 1.** (Continued)

| Variables | Definition | Mean | Standard Deviation | Minimum | Maxi-mum |
|---|---|---|---|---|---|
| *Province:* | | | | | |
| Punjab | 1 if the household belong to Punjab, 0 otherwise | 0.59 | 0.49 | 0 | 1 |
| Sindh | 1 if the household belong to Sindh, 0 otherwise | 0.35 | 0.48 | 0 | 1 |
| Baluchistan | 1 if the household belong to Baluchistan, 0 otherwise | 0.045 | 0.13 | 0 | 1 |
| Observations | Number of households | 16155 | | | |

*Source*: Estimated by authors based on PSLM data for the year 2019−20.

least square can produce inefficient estimators. In other words, in the presence of outlier, the variances of OLS estimators are not provided by the usual OLS formulas. But if we persist in using the usual OLS formulas, the t and F tests based on them can be highly misleading, resulting in erroneous conclusions. Various tests are used for checking heteroscedasticity in cross-sectional data, among these test Breusch-Pagan/Cook-Weisberg test is considered superior than other tests. Thus, we applied this test for checking heteroskedasticity in our data set. The null hypothesis of the test is there is no heteroscedasticity in the data. If the value of $\chi^2$ (Chi square) statistics of Breusch-Pagan/Cook-Weisberg test is significant reject the null hypothesis [41].

In order to avoid the problem of outlier we estimate regression (6) use the method of robust least square (RLS). RLS offers an alternative to OLS regression that is less sensitive to outlier and still defines a linear relationship between the outcome and the predictors. RLS works by first fitting the OLS regression model from above and identifying the records that have a Cook's distance greater than 1. Then, a regression is run in which those records with Cook's distance greater than 1 are given zero weight. From this model, weights are assigned to records according to the absolute difference between the predicted and actual values (the absolute residual). The records with small absolute residuals are weighted more heavily than the records with large absolute residuals. Then, another regression is run using these newly assigned weights, and then new weights are generated from this regression. This process of regressing and reweighting is iterated until the differences in weights before and after a regression is sufficiently close to zero [42].

## Results and discussion

### Existing solid waste collection and management system available to households

From Fig 4 we observed that in Pakistan the average monthly household income is PKR 25497 (Approximately USD 1; 1USD = PKR 220). Households from Baluchistan have the highest average monthly income of PKR 28614, followed by the households in Khyber Pakhtunkhwa (PKR 28356), households in Punjab (PKR 25716), and households in Sindh (PKR 24574). Similarly, the monthly income of the rural household is higher (PKR 27546) than that of the monthly income of the urban households (PKR 25264) as shown in Fig 5. This differences in the average income of rural areas compared to urban areas can be attributed mainly to two factors. First, rural incomes benefit from land ownership, agriculture, and communal resources while residents of urban areas rely more on wage-based income. Second, the cost of living in rural areas is lower, making the rural residents income stretch further while urban areas face higher expenses, making the urban residents income rigid. However, this result is in contradiction with the outcomes of Khalid [43], he found that household monthly income in Pakistan is higher for urban households in comparison to rural households. However, instead of using PSLM 2019–2020 data he used Household Integrated Expenditure survey (HIES-2018–2019).

Punjab is most populous province and that is why sample size is large however, Baluchistan is biggest in area but least populous province. Conversely, households from Baluchistan on average pay the highest money of PKR 382 per month for waste collection and disposal services, followed by the households in Khyber Pakhtunkhwa (PKR 263 per month),

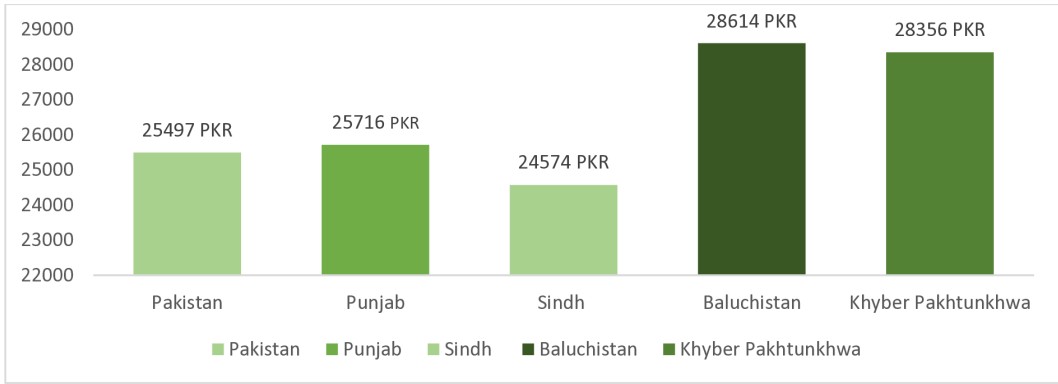

*Source*: Estimated by authors based on PSLM data for the year 2019-20.

**Fig 4. Province-wise household mean income (PKR/Month).**

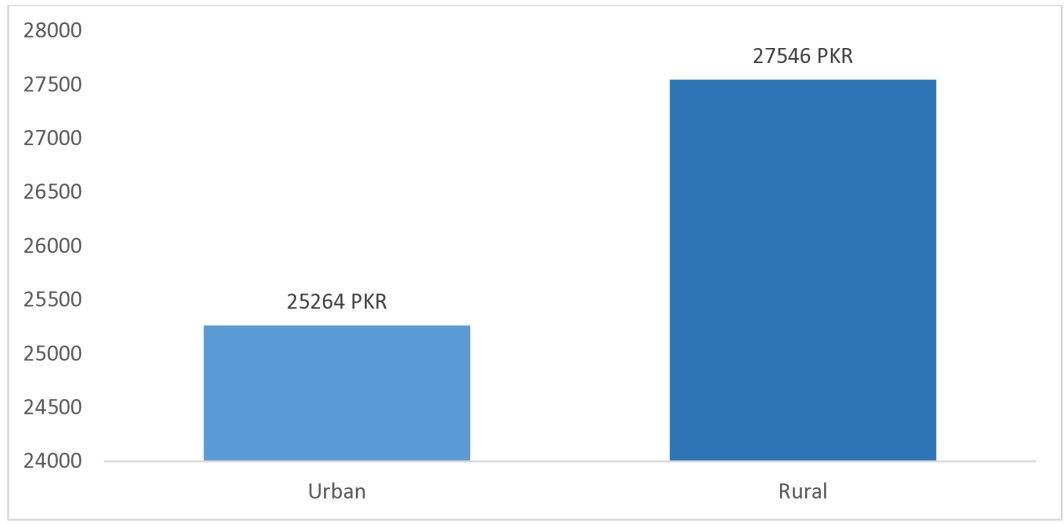

*Source*: Estimated by authors based on PSLM data for the year 2019-20.

**Fig 5. Region-wise household mean income (PKR/Month).**

households in Sindh (PKR 228 per month), and households in Punjab (PKR 191 per month) as shown in Fig 6. Similarly, from Fig 7, it is observed that the rural household pay more cash (PKR 308 per month) for waste collection and disposal services as compared to the urban households (PKR 203 per month). Furthermore, from Fig 6 we observe that in Pakistan on average the households pay PKR 214 per month for waste collection and disposal services. Current solid waste management system in the country is poor and needs more improvement.

The results can have two possible interpretations. Firstly, wealthier people pay more cash payment for solid waste management. For instance, Baluchistan has higher average income and thus pays more for solid waste management system. Similarly, rural residents have higher mean income and thus they spend more for garbage collection as compared to urban counterparts. It could be because there is well established system of waste management in urban areas provided

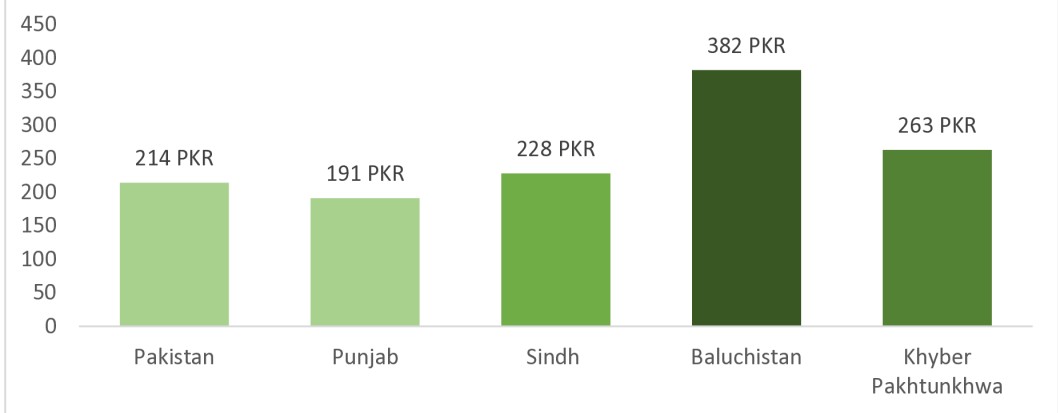

*Source*: Estimated by authors based on PSLM data for the year 2019-20.

**Fig 6. Province-wise household mean cash payment for waste collection and disposal services (PKR/Month).**

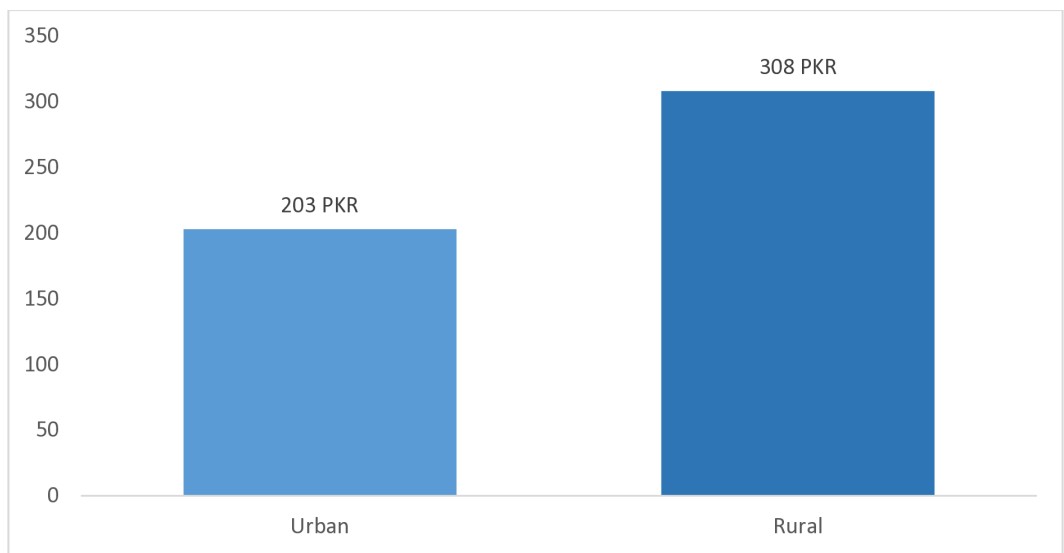

*Source*: Estimated by authors based on PSLM data for the year 2019-20.

**Fig 7. Region-wise household mean cash payment for waste collection and disposal services (PKR/Month).**

by government and hence urban residents spend less money for waste collection. On the other hands, waste collection infrastructure is very poor in rural areas and therefore rural people spend more money for waste disposal. Secondly, waste management service is important, so people spend money for it despite to their financially soundness. For instance, Punjab province has lower mean income than Sindh but still former pays more than later. Akhtar et al. [44] found that people willingness to pay regarding waste management is USD 4.8 per month in Pakistan. Therefore, study recommends that for increasing the cash payment for waste collection and disposal services the waste collection authorities must provide better and modern solid waste management system to the households. The upgradation of the existing solid waste management system can also increase the households cash payments for waste collection and disposal services.

From Fig 8, it is observed that out of 16155 households, waste from 72 percent of households has been collected by private van/cart from door step whereas waste from 28 percent of households has been collected by municipality van from door step. This show the municipality role in collecting the waste from the households is still limited in the country. The municipal corporations in the big cities are under the control of city government whereas in small town and rural setup it works under the tehsil governments. However, in the country the local government system is not so powerful, thus the municipalities get a small budget and resultantly they are unable to reach to every corner of the city or to each rural area. Thus, they are collecting a small portion of waste from the households and the remaining households are bound to dispose their waste using private van or carts.

In the previous Fig 8, we observed that waste from majority of the households included in the analysis has been collected by private van/cart from door step but in the current scenario it does not mean door-to-door waste collection services. In many areas of the country municipal corporations in the big cities installed public bin or identified waste collection points. Similarly, the private waste collectors also identified waste collection point near to door steps. Thus, waste collection points and disposal bin are available/accessible to 83 percent of the households whereas waste collection point near the door step and public bin is not available/accessible to 17 percent of the households as shown in Fig 9. The non-availability of public bin and waste collection point facilities particularly in rural areas of the country, exacerbates environmental and public health concerns, as improper disposal leads to waste accumulation in open spaces or water bodies. This disparity underscores the need for more comprehensive waste management systems, especially in rural areas.

Fig 10 shows that the 52 percent of the household spent 1–5 minutes on a round trip to the nearest public bin/waste collection point. Similarly, 19 percent of the household spent 6–10 minutes, 8 percent of the households spent 11–15 minutes, 3 percent of the households spent 16–20 minutes, 1 percent of the households spent 21–25 minutes, and 1 percent of the household spent 26 and more minutes on a round trip to the nearest public bin/collection point, respectively. Though there is no universally established recommended time for households to reach the nearest public bin/waste collection point. However, UN-Habitat [45] suggested that waste bin/collection point should be within a walking distance of under 5–10 minutes. If the time taken to reach a waste collection point exceeds 10–15 minutes, households are more likely to

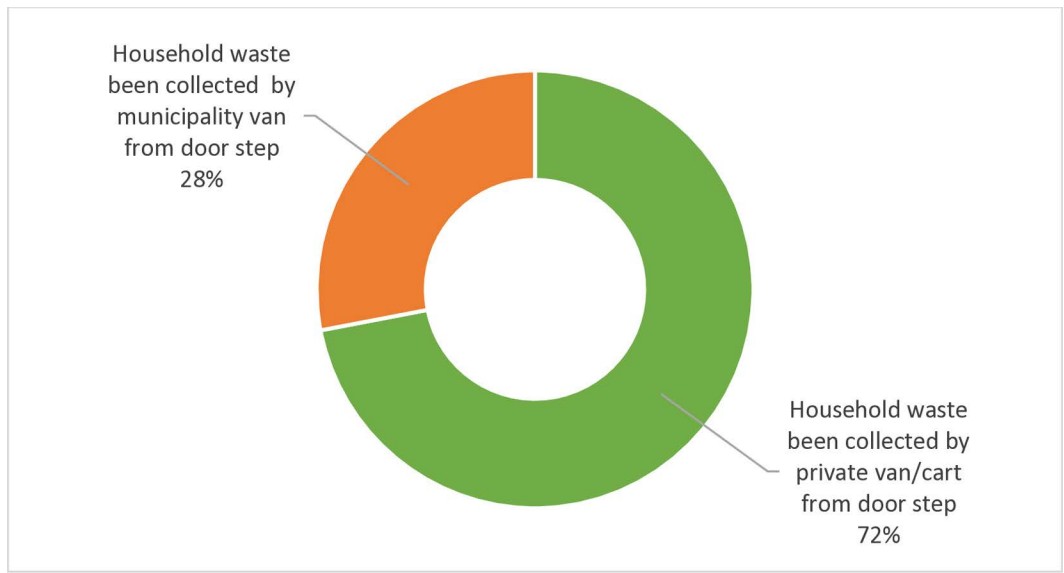

*Source*: Estimated by authors based on PSLM data for the year 2019-20.

**Fig 8. Households waste collection.**

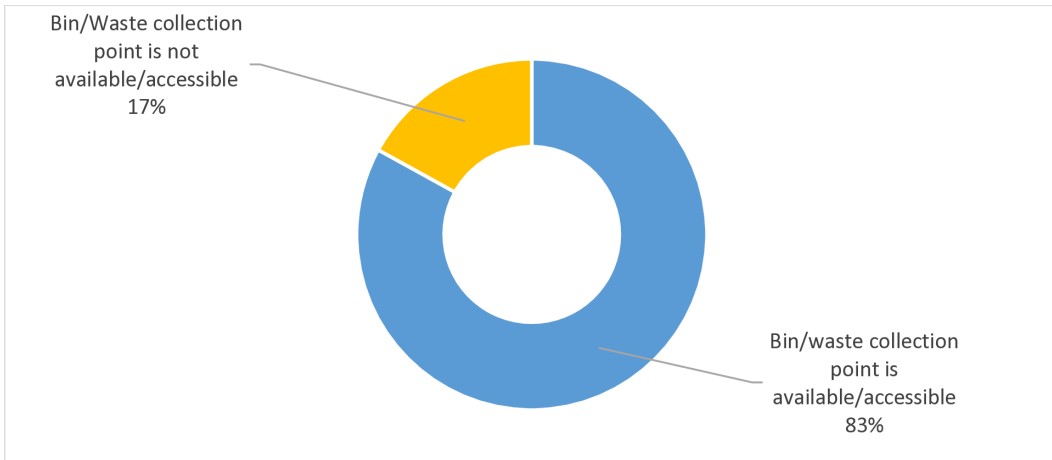

*Source*: Estimated by authors based on PSLM data for the year 2019-20.

**Fig 9. Bin/ Waste collection point availability/accessibility to households.**

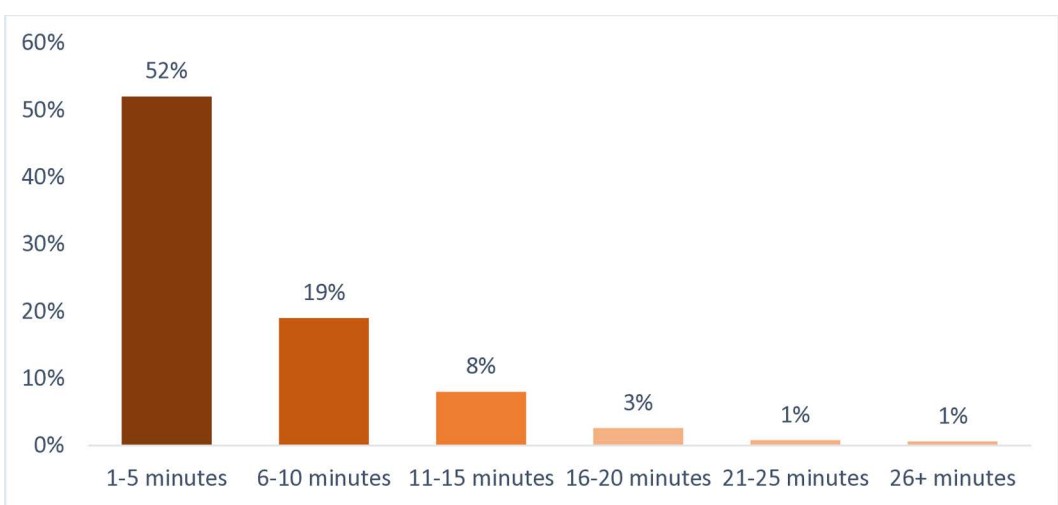

*Source*: Estimated by authors based on PSLM data for the year 2019-20.

**Fig 10. Average time the household spent on a round trip to the nearest public bin/waste collection point.**

engage in improper waste disposal. Keeping this point in mind, in Pakistan, average disposal times fall within the recommended range.

In Fig 11, 42 percent of the households reported that the nearest public bin/waste collection point available to them is emptied/cleared by the concern person(s) or local authority(s) every day. In similar fashion, 11 percent of the households reported this duration once a week, 6 percent of the households reported this duration twice a week, and only 3 percent of household reported this duration thrice a week, respectively. On the other hand, 38 percent of the household don't know about the nearest public bin emptied duration. This is a good sign that in the country for waste disposal bin/collection points is available to majority of households. However, the real problem is that these bins/collection points are far away from the house site and the households spent more time to reach to bin/collection points for waste disposing. Therefore,

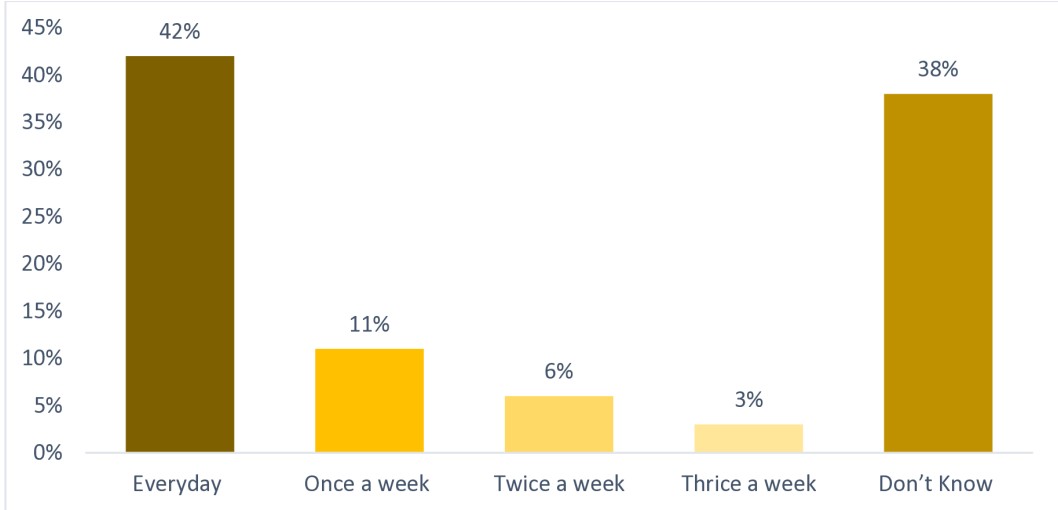

*Source*: Estimated by authors based on PSLM data for the year 2019-20.

**Fig 11. Duration of the nearest public bin/waste collection point emptied/cleared.**

it is recommended that the authorities put the bin/collection points near to the households' residence that they are dispose their waste in lesser time and lesser efforts. The second problem, is these bins/collection points are emptied/cleared by the concern person(s) or local authority(s) after few days. If the bin/collection point is not emptied on time than it can cause various diseases and can adversely affect the households' health. Therefore, it is recommended that the authorities must devise proper and quick method for clearing the bin/collection points in the recommended time period.

## Descriptive statistics

Descriptive statistics and the definition of the important variables used in this study is given in Table 1. It is observed that on average the Pakistani households pay PKR 214 per month for waste collection and disposal services. Contrasting the explanatory variables, we find that the average monthly household income in Pakistan is PKR 25497. Comparing household socio-demographic characteristics, it is observed that the average education level of the household head in the country is 9 years, while the average age of the household head is 48 years. Similarly, only 8 percent of households have a female head. On average, about one person in each household earns income for his/her household. Comparing the Household waste management characteristics, it is observed that from 72 percent of households' waste has been collected by private van/cart from door step. For waste collection and disposal bin/waste collection point is available/accessible to 17 percent of the households. Furthermore, 52 percent of the household spent 1–5 minutes on a round trip to the nearest public bin/collection point. Similarly, 19 percent of the household spent 6–10 minutes, 8 percent of the households spent 11–15 minutes, 3 percent of the households spent 16–20 minutes, only 0.8 percent of the households spent 21–25 minutes on a round trip to the nearest public bin/collection point, respectively. 42 percent of the households reported that the nearest public bin/collection point available to them is emptied/cleared by the concern person(s) or local authority(s) every day. In similar fashion, 11 percent of the households reported this duration once a week, 6 percent of the households reported this duration twice a week, and only 3 percent of household reported this duration thrice a week, respectively. Comparing the household housing characteristics, it is observed that 83 percent of the households living in their own houses whereas 10 percent of the households living in rented houses. Moreover, the households are living on average in 2 rooms houses. Similarly, 77 percent of the households living in independent houses, 13 percent of the

households living in part of large units, 7 percent of the households living in part of compounds, and only 3 percent of households living in apartments. Furthermore, 90 percent of the household living in urban areas. Finally, 59 percent of the households living in Punjab, 36 percent of the households living in Sindh, and 4.5 living in Baluchistan.

## Factors affecting the households cash payments for waste collection and disposal services

The results of the estimated regressions for Pakistan are given in Table 2, while the results of the estimated regressions for the four provinces are given in Table 3 (Punjab, Sindh, Khyber Pakhtunkhwa, Baluchistan). The regressions are estimated via OLS and RSL and given in separate columns in each Table. The results of the diagnostic check (R squared, F statistics and Root MSE) of all the regressions are reported in the last panel of Table 2 and Table 3, respectively. In our seven Regression models estimated through OLS have an R-squared ranging from 0.08 to 0.29, while the R-squared values in regression models estimated through RLS are relatively higher, ranging from 0.12 to 0.31. This indicates that the explanatory power of the RLS regressions is greater than that of the OLS regressions. The results of the F-statistics show that all regression models are statistically significant at the 1 percent level, indicating that the explanatory variables have a significant impact on the dependent variable. The root mean square error (Root MSE) of the seven regressions estimated via OLS ranges from 0.50 to 0.65, suggesting a reasonable estimation accuracy. Based on the statistically significant values of the Breusch-Pagan/Cook-Weisberg test for heteroskedasticity in OLS regressions for models 1–6, we reject the null hypothesis of homoscedasticity, indicating the presence of heteroskedasticity in the data. However, for model 7, the test for heteroskedasticity yields an insignificant result, leading us to accept the null hypothesis of no heteroskedasticity. This is likely due to the smaller sample size used in model 7, which reduces the likelihood of outliers. Since heteroskedasticity is present in the six regression models, therefore; we proceeded with robust least squares (RLS) regression to address the issue caused by outliers.

From Table 2 (Pakistan and urban), we found statically significant negative impact of income on the households' cash payment for solid waste collection and disposal services. This shows that an increase in household income brings a significant decrease in their cash payments for waste collection and disposal services. These results go in line with a priori expectation given that prevailing differences in the attitudes of households towards waste collection and disposal services are often borne out of their income and poverty levels. In the developing country like Pakistan, households are mostly poor and unable to independently support their living expenses. Therefore, when their income increases, they increase their expenses on other essentials like food items, fuel, utilities, footwear, clothing, housing, education, and health while the increase in their incomes negatively affect their payment on waste collection and disposal. According to Akmal & Jamil [46] with limited income, households often need to allocate most of their money to essential needs like food and shelter, leaving little left over for solid waste collection and disposal services. For Sindh (Table 3), we find similar result. For rural areas in Pakistan (Table 2), Punjab, Khyber Pakhtunkhwa, and Baluchistan, we found statically insignificant impact of income on the households' cash payment for solid waste collection and disposal services. In related studies, Oyekale [18] and Omotayo et al. [1] found that household's income positively affecting their payment for waste disposal. In other studies, researchers found that households' income positively affects their willingness to pay for solid waste management system and services [4,31,33,36].

Education of household head can promote environmental awareness and conservation behaviors. This is as a result of the expectation that educated people would have better values for environmental conservation. The estimated coefficients of education of household head for Pakistan (both urban and rural areas) and for the four provinces are positive and statistically significant. These indicate that households that have educated heads are paying more for waste collection and disposal services. This implies that increase in the household head education increases the household knowledge about the short-term and long-term impacts of accumulated solid waste on human health and esthetic value of the environment which increases the households' payment for solid waste collection and disposal services. This is because educated household head clearly understand all the threats, diseases and damages caused by improper solid waste collection

**Table 2. Results of ordinary least square and robust least square regressions for Pakistan.**

| Model: | 1 | | 2 | | 3 | |
|---|---|---|---|---|---|---|
| | Pakistan | | Urban | | Rural | |
| | OLS | RLS | OLS | RLS | OLS | RLS |
| *Dependent variable: Household cash payment for waste collection and disposal services (ln)* | | | | | | |
| *Household income:* | | | | | | |
| Income (ln) | −0.00846 | −0.0123** | −0.00709 | −0.0132** | −0.0314 | −0.00782 |
| | (0.00593) | (0.00509) | (0.00598) | (0.00534) | (0.0238) | (0.0166) |
| *Household socio-demographic characteristics:* | | | | | | |
| Education of household head | 0.0223*** | 0.0219*** | 0.0216*** | 0.0210*** | 0.0272*** | 0.0242*** |
| | (0.000846) | (0.000727) | (0.000856) | (0.000765) | (0.00343) | (0.00239) |
| Age of the household head | 0.00406*** | 0.00402*** | 0.00419*** | 0.00409*** | 0.00177 | 0.00224** |
| | (0.000341) | (0.000293) | (0.000345) | (0.000308) | (0.00136) | (0.000947) |
| Male household head (Reference) | | | | | | |
| Female household head (Yes=1, No=0) | 0.0345** | 0.0260* | 0.0507*** | 0.0328** | −0.0982* | −0.0706* |
| | (0.0161) | (0.0138) | (0.0165) | (0.0147) | (0.0574) | (0.0401) |
| Number of income earners in household | −0.0118** | −0.00868* | −0.00987 | −0.00845 | −0.0238 | −0.00448 |
| | (0.00594) | (0.00510) | (0.00600) | (0.00536) | (0.0231) | (0.0161) |
| *Household waste management techniques:* | | | | | | |
| Household waste been collected by municipality van from door step (Reference) | | | | | | |
| Household waste been collected by private van/ cart from door step (Yes=1, No=0) | 0.0762*** | 0.0852*** | 0.0862*** | 0.0847*** | −0.220*** | 0.0426* |
| | (0.00519) | (0.00446) | (0.00508) | (0.00454) | (0.0339) | (0.0237) |
| Bin/collection point available/accessible (Yes=1, No=0) | −0.0988* | −0.121** | −0.119** | −0.116** | −0.126 | −0.0257 |
| | (0.0584) | (0.0502) | (0.0584) | (0.0522) | (0.254) | (0.177) |
| *Average time the household spent on a round trip to the nearest public bin/collection point:* | | | | | | |
| 26+minutes (Reference) | | | | | | |
| 1–5 minutes (Yes=1, No=0) | 0.196*** | 0.116** | 0.161*** | 0.109** | 0.531** | 0.0928 |
| | (0.0580) | (0.0498) | (0.0580) | (0.0518) | (0.252) | (0.176) |
| 6–10 minutes (Yes=1, No=0) | 0.221*** | 0.159*** | 0.197*** | 0.149*** | 0.442* | 0.178 |
| | (0.0583) | (0.0501) | (0.0582) | (0.0520) | (0.255) | (0.178) |
| 11–15 minutes (Yes=1, No=0) | 0.229*** | 0.156*** | 0.211*** | 0.151*** | 0.323 | 0.0677 |
| | (0.0594) | (0.0510) | (0.0592) | (0.0529) | (0.266) | (0.186) |
| 16–20 minutes (Yes=1, No=0) | 0.238*** | 0.165*** | 0.226*** | 0.153*** | 0.102 | 0.199 |
| | (0.0634) | (0.0544) | (0.0629) | (0.0562) | (0.328) | (0.229) |
| 21–25 minutes (Yes=1, No=0) | 0.212*** | 0.183*** | 0.185** | 0.177*** | 0.235 | 0.221 |
| | (0.0749) | (0.0643) | (0.0737) | (0.0659) | (0.477) | (0.333) |
| *Duration of the nearest public bin/collection point emptied/cleared:* | | | | | | |
| Don't Know (Reference) | | | | | | |
| Every day (Yes=1, No=0) | −0.0451*** | 0.00698 | −0.0269** | 0.00317 | −0.159*** | 0.0597 |
| | (0.0128) | (0.0110) | (0.0129) | (0.0115) | (0.0544) | (0.0380) |
| Once a week (Yes=1, No=0) | −0.0336** | 0.00177 | −0.00417 | 0.0205 | −0.322*** | −0.267*** |
| | (0.0163) | (0.0140) | (0.0164) | (0.0146) | (0.0720) | (0.0503) |
| Twice a week (Yes=1, No=0) | 0.00220 | 0.0774*** | 0.0455** | 0.0973*** | −0.305*** | −0.166*** |
| | (0.0201) | (0.0172) | (0.0202) | (0.0181) | (0.0839) | (0.0586) |
| Thrice a week (Yes=1, No=0) | 0.161*** | 0.102*** | 0.209*** | 0.116*** | −0.163 | −0.124 |

*(Continued)*

**Table 2.** (Continued)

| Model: | 1 | | 2 | | 3 | |
|---|---|---|---|---|---|---|
| | (0.0276) | (0.0237) | (0.0275) | (0.0246) | (0.122) | (0.0855) |
| *Household housing characteristics:* | | | | | | |
| Rent free (Reference) | | | | | | |
| Own house (Yes = 1, No = 0) | 0.0328* | 0.0266* | 0.0430** | 0.0289* | −0.0763 | 0.0145 |
| | (0.0175) | (0.0150) | (0.0175) | (0.0157) | (0.0716) | (0.0500) |
| On rent (Yes = 1, No = 0) | −0.0147 | −0.0162 | −0.00592 | −0.0145 | −0.0773 | 0.00160 |
| | (0.0218) | (0.0187) | (0.0220) | (0.0197) | (0.0856) | (0.0598) |
| Number of rooms in the house | −0.00323 | −0.00386 | 0.000153 | −0.00128 | −0.0236* | −0.0209** |
| | (0.00337) | (0.00290) | (0.00340) | (0.00304) | (0.0136) | (0.00953) |
| *Dwelling type:* | | | | | | |
| Other dwelling type (Reference) | | | | | | |
| Independent house (Yes = 1, No = 0) | −0.182** | −0.184*** | −0.159** | −0.187*** | −0.400 | −0.181 |
| | (0.0788) | (0.0676) | (0.0793) | (0.0708) | (0.316) | (0.220) |
| Apartment (Yes = 1, No = 0) | −0.240*** | −0.241*** | −0.228*** | −0.253*** | −0.264 | −0.0809 |
| | (0.0825) | (0.0709) | (0.0829) | (0.0741) | (0.347) | (0.243) |
| Part of the large unit (Yes = 1, No = 0) | −0.174** | −0.183*** | −0.166** | −0.187*** | −0.300 | −0.163 |
| | (0.0794) | (0.0682) | (0.0800) | (0.0715) | (0.318) | (0.222) |
| Part of compound (Yes = 1, No = 0) | −0.184** | −0.174** | −0.159** | −0.181** | −0.357 | −0.0746 |
| | (0.0803) | (0.0689) | (0.0807) | (0.0721) | (0.328) | (0.229) |
| *Region:* | | | | | | |
| Rural (Reference) | | | | | | |
| Urban (Yes = 1, No = 0) | −0.0887*** | −0.0410*** | | | | |
| | (0.00764) | (0.00656) | | | | |
| *Province:* | | | | | | |
| Khyber Pakhtunkhwa (Reference) | | | | | | |
| Punjab (Yes = 1, No = 0) | −0.314*** | −0.316*** | −0.298*** | −0.329*** | −0.319*** | −0.202*** |
| | (0.0340) | (0.0292) | (0.0364) | (0.0326) | (0.0973) | (0.0679) |
| Sindh (Yes = 1, No = 0) | −0.141*** | −0.153*** | −0.127*** | −0.159*** | −0.786*** | −0.329*** |
| | (0.0346) | (0.0297) | (0.0368) | (0.0329) | (0.131) | (0.0917) |
| Baluchistan (Yes = 1, No = 0) | −0.00118 | −0.0368*** | −0.0193* | −0.0284*** | 0.0429 | −0.0402** |
| | (0.00991) | (0.00851) | (0.0109) | (0.00972) | (0.0271) | (0.0189) |
| Constant | 5.221*** | 5.191*** | 4.988*** | 5.127*** | 6.325*** | 5.221*** |
| | (0.106) | (0.0914) | (0.107) | (0.0959) | (0.425) | (0.297) |
| *Diagnostic check:* | | | | | | |
| Observations | | 16,155 | 14,505 | 14,505 | 1,650 | 1,650 |
| R-squared | 0.12 | 0.14 | 0.13 | 0.15 | 0.16 | 0.12 |
| F Statistics | 83.6*** | 97.5*** | 83.3*** | 97.8*** | 11.8*** | 8.7*** |
| Root MSE | 0.55 | | 0.53 | | 0.70 | |
| Breusch-Pagan/ Cook-Weisberg test for heteroskedasticity | 122.5*** | | 11.5** | | 253.6*** | |

*Source*: Estimated by authors based on PSLM data for the year 2019−20. Note: Robust standard errors in parentheses,

\***p < 0.01,

\**p < 0.05,

\*p < 0.1.

**Table 3. Results of ordinary least square and robust least square regressions for four provinces of Pakistan.**

| Model: | 4 | | 5 | | 6 | | 7 | |
|---|---|---|---|---|---|---|---|---|
| Province: | Punjab | | Sindh | | Baluchistan | | Khyber Pakhtunkhwa | |
| | OLS | RLS | OLS | RLS | OLS | RLS | OLS | RLS |
| *Dependent variable: Household cash payment for waste collection and disposal services (ln)* | | | | | | | | |
| *Household income:* | | | | | | | | |
| Income (ln) | 0.00962 | 0.00323 | −0.0288*** | −0.0271*** | −0.0151 | −0.00259 | 0.0464 | 0.0513 |
| | (0.00785) | (0.00685) | (0.00914) | (0.00814) | (0.0316) | (0.0155) | (0.0495) | (0.0493) |
| *Household socio-demographic characteristics:* | | | | | | | | |
| Education of household head | 0.0205*** | 0.0193*** | 0.0271*** | 0.0265*** | 0.00775* | 0.00876*** | 0.0149** | 0.0150** |
| | (0.00111) | (0.000971) | (0.00135) | (0.00120) | (0.00456) | (0.00223) | (0.00620) | (0.00617) |
| Age of the household head | 0.00324*** | 0.00336*** | 0.00591*** | 0.00575*** | −0.00164 | −0.000942 | 0.00439 | 0.00220 |
| | (0.000444) | (0.000388) | (0.000546) | (0.000485) | (0.00196) | (0.000958) | (0.00291) | (0.00290) |
| Male household head (Reference) | | | | | | | | |
| Female household head (Yes = 1, No = 0) | 0.0286 | 0.0156 | 0.0724*** | 0.0509** | −0.181 | −0.0517 | −0.0809 | −0.0613 |
| | (0.0197) | (0.0172) | (0.0274) | (0.0244) | (0.122) | (0.0596) | (0.151) | (0.150) |
| Number of income earners in household | −0.0140* | −0.00534 | 0.00679 | −0.0110 | 0.0204 | 0.0284* | −0.0849** | −0.120*** |
| | (0.00726) | (0.00634) | (0.0109) | (0.00972) | (0.0312) | (0.0153) | (0.0397) | (0.0395) |
| *Household solid waste management techniques:* | | | | | | | | |
| Household waste been collected by municipality van from door step (Reference) | | | | | | | | |
| Household waste been collected by private van/cart from door step (Yes = 1, No = 0) | 0.0906*** | 0.101*** | 0.0654*** | 0.0619*** | 0.106 | 0.0981*** | −0.107** | −0.0894** |
| | (0.00630) | (0.00550) | (0.00963) | (0.00857) | (0.0667) | (0.0326) | (0.0411) | (0.0409) |
| Bin/collection point available/accessible (Yes = 1, No = 0) | −0.0246 | −0.0139 | −0.206** | −0.210*** | 0.686** | 0.369** | −1.890*** | −1.935*** |
| | (0.0824) | (0.0720) | (0.0806) | (0.0717) | (0.344) | (0.169) | (0.592) | (0.590) |
| *Average time the household spent on a round trip to the nearest public bin/collection point:* | | | | | | | | |
| 26 + minutes (Reference) | | | | | | | | |
| 1–5 minutes (Yes = 1, No = 0) | 0.0402 | −0.0222 | 0.278*** | 0.215*** | 0.377 | 0.0301 | 1.656*** | 1.648*** |
| | (0.0816) | (0.0713) | (0.0805) | (0.0716) | (0.342) | (0.167) | (0.580) | (0.578) |
| 6–10 minutes (Yes = 1, No = 0) | 0.0425 | −0.0206 | 0.388*** | 0.324*** | −0.00771 | 0.111 | 1.513** | 1.525*** |
| | (0.0822) | (0.0718) | (0.0804) | (0.0715) | (0.346) | (0.169) | (0.583) | (0.581) |
| 11–15 minutes (Yes = 1, No = 0) | 0.132 | 0.0565 | 0.322*** | 0.236*** | −0.203 | −0.0499 | 1.878*** | 1.873*** |
| | (0.0838) | (0.0732) | (0.0819) | (0.0728) | (0.349) | (0.171) | (0.619) | (0.616) |
| 16–20 minutes (Yes = 1, No = 0) | 0.194** | 0.107 | 0.120 | 0.0678 | 0.0184 | −0.0190 | 2.082** | |
| | (0.0877) | (0.0766) | (0.0918) | (0.0817) | (0.360) | (0.176) | (0.834) | |
| 21–25 minutes (Yes = 1, No = 0) | 0.0929 | 0.0876 | 0.241** | 0.195* | 0.0240 | 0.104 | 1.625** | |
| | (0.0985) | (0.0861) | (0.119) | (0.106) | (0.580) | (0.284) | (0.822) | |
| *Duration of the nearest public bin/collection point emptied/cleared:* | | | | | | | | |
| Don't Know (Reference) | | | | | | | | |
| Every day (Yes = 1, No = 0) | −0.0180 | −0.0186 | 0.0377* | 0.0776*** | −0.609*** | −0.145*** | −0.193* | −0.155 |
| | (0.0174) | (0.0152) | (0.0194) | (0.0173) | (0.0950) | (0.0465) | (0.116) | (0.116) |
| Once a week (Yes = 1, No = 0) | 0.0142 | −0.0354* | −0.0339 | 0.0166 | −0.643*** | −0.247*** | 0.260* | 0.383*** |
| | (0.0239) | (0.0209) | (0.0225) | (0.0200) | (0.0857) | (0.0419) | (0.140) | (0.139) |
| Twice a week (Yes = 1, No = 0) | 0.0572** | 0.0917*** | −0.0172 | 0.0287 | −0.315*** | −0.105** | −0.191 | −0.0847 |
| | (0.0273) | (0.0239) | (0.0308) | (0.0274) | (0.0993) | (0.0486) | (0.182) | (0.181) |

*(Continued)*

| Model: | 4 | | 5 | | 6 | | 7 | |
|---|---|---|---|---|---|---|---|---|
| Thrice a week (Yes = 1, No = 0) | 0.261*** | 0.147*** | 0.0367 | −0.0336 | −0.299* | −0.141* | 0.122 | 0.146 |
| | (0.0334) | (0.0292) | (0.0540) | (0.0480) | (0.167) | (0.0820) | (0.358) | (0.357) |
| *Household housing characteristics:* | | | | | | | | |
| Rent free (Reference) | | | | | | | | |
| Own house (Yes = 1, No = 0) | 0.0619*** | 0.0460** | 0.00936 | 0.0172 | −0.121 | 0.0584 | −0.142 | −0.175 |
| | (0.0234) | (0.0204) | (0.0267) | (0.0237) | (0.0949) | (0.0465) | (0.131) | (0.131) |
| On rent (Yes = 1, No = 0) | 0.0444 | 0.0194 | −0.0716** | −0.0217 | −0.285** | −0.0214 | −0.112 | −0.162 |
| | (0.0283) | (0.0247) | (0.0348) | (0.0310) | (0.128) | (0.0629) | (0.165) | (0.164) |
| Number of rooms in the house | 0.00373 | −0.000632 | −0.00862 | −0.000919 | −0.00893 | −0.0145 | 0.0562* | 0.0243 |
| | (0.00440) | (0.00385) | (0.00526) | (0.00468) | (0.0196) | (0.00958) | (0.0316) | (0.0315) |
| *Dwelling type:* | | | | | | | | |
| Other dwelling type (Reference) | | | | | | | | |
| Independent house (Yes = 1, No = 0) | −0.156 | −0.151 | −0.203* | −0.244** | 0.0218 | −0.0666 | −0.579 | −0.489 |
| | (0.105) | (0.0921) | (0.112) | (0.0993) | (0.217) | (0.106) | (0.589) | (0.586) |
| Apartment (Yes = 1, No = 0) | −0.290*** | −0.299*** | −0.113 | −0.147 | | | −0.261 | −0.170 |
| | (0.110) | (0.0958) | (0.119) | (0.106) | | | (0.597) | (0.595) |
| Part of the large unit (Yes = 1, No = 0) | −0.136 | −0.124 | −0.248** | −0.288*** | 0.180 | −0.0207 | −0.284 | −0.182 |
| | (0.106) | (0.0928) | (0.113) | (0.100) | (0.221) | (0.108) | (0.600) | (0.597) |
| Part of compound (Yes = 1, No = 0) | −0.173 | −0.158* | −0.168 | −0.194* | 0.158 | 0.0712 | −0.842 | −0.708 |
| | (0.107) | (0.0938) | (0.114) | (0.102) | (0.242) | (0.119) | (0.620) | (0.618) |
| *Region:* | | | | | | | | |
| Rural (Reference) | | | | | | | | |
| Urban (Yes = 1, No = 0) | −0.0721*** | −0.0503*** | 0.0232 | 0.0205 | −0.0378 | 0.116*** | −0.212*** | −0.233*** |
| | (0.00868) | (0.00759) | (0.0317) | (0.0282) | (0.0314) | (0.0154) | (0.0520) | (0.0518) |
| Constant | 4.706*** | 4.748*** | 4.953*** | 4.995*** | 5.130*** | 4.672*** | 5.894*** | 6.010*** |
| | (0.135) | (0.118) | (0.161) | (0.144) | (0.428) | (0.210) | (0.816) | (0.812) |
| *Diagnostic check:* | | | | | | | | |
| Observations | 9,555 | 9,555 | 5,592 | 5,592 | 733 | 733 | 275 | 273 |
| R-squared | 0.08 | 0.11 | 0.13 | 0.15 | 0.29 | 0.30 | 0.29 | 0.31 |
| F Statistics | 38.1*** | 47.1*** | 34.2*** | 39.5*** | 13*** | 13.4*** | 4.4*** | 5.1*** |
| Root MSE | 0.55 | | 0.50 | | 0.65 | | 0.56 | |
| Breusch-Pagan/ Cook-Weisberg test for heteroskedasticity | 55.6*** | | 36.2*** | | 441.5*** | | 0.82 | |

*Source*: Estimated by authors based on PSLM data for the year 2019−20. Note: Robust standard errors in parentheses,

***p < 0.01,

**p < 0.05,

*p < 0.1.

and disposal [4]. Therefore, household heads with higher education may ensure that their household wastes are properly collected and disposed [35]. In some previous studies, Olukanni et al. [34], Omotayo et al. [1], and Oyekale [18] found positive relationship between household head education and payments for waste collection and disposal services. Similarly, literature also found a significant positive impact of household head education and willingness to pay for improved solid waste disposal services [22,31,36–40,44,47–51].

Older and experienced citizens are more likely to be aware of the adverse impact of improper disposal and management of solid waste on public health and the natural environment. Consequently, households with older heads tend to pay more for waste disposal and collection services, as observed in Pakistan (both urban and rural areas) and its two provinces, Punjab, and Sindh. This can be explained by the fact that as people become older, they tend to understand the need for a clean environment more compared to younger people. Older people tend to appreciate the importance and need for a clean and safe environment hence a positive relationship between age and cash payment for waste disposal and collection services [22]. This result is in line with findings of previous studies [1,18,31,37–39].

According to the tradition in Pakistan one of the roles of female is to keep the house clean and dispose of the waste. Thus, it is expected that female will more likely pay for improved solid waste management services as compared to male. The female headed household shows a significant positive impact on household cash payment for waste collection and disposal services in case of Pakistan, urban areas, and Sindh. This indicates that female headed households are pay more for solid waste collection and disposal services than males, a situation that can be explained by the fact that in Pakistan, urban areas of the country, and in Sindh province women are traditionally responsible for maintaining hygiene and sanitation in the home and cleaning and waste disposal activities. This result lends credence to findings of the previous research [23,33,38,39,50] but in contradiction with the findings Mulat et al. [36]. However, for rural areas in Pakistan, the female headed household shows a significant negative impact on household cash payment for waste collection and disposal services. This may be due to financial constraints make it difficult for female headed households to allocate funds for waste management services. The reliance on traditional waste disposal methods such as burning, burying, or self-disposal of waste may be another reason for this negative result. Lower awareness regarding formal waste management services may be another possible reason.

For Pakistan, Baluchistan, and Khyber Pakhtunkhwa, we found statically significant negative impact of number of income earners on the households' cash payment for solid waste collection and disposal services. This shows that an increase in number of income earner in the household brings a significant decrease in their cash payments for waste collection and disposal services. As the number of income earners in a household increase, the total household income may rise, but the per capita disposable income might not significantly increase, especially in low-income households. Households with multiple earners often allocate their income toward essential needs such as food, healthcare, and education, deprioritizing waste management expenditures. Moreover, in developing countries like Pakistan, low-income households tend to rely more on informal or self-managed waste disposal methods [52]. Therefore, an increase in number of earners in the households negatively affect their payment on waste collection and disposal services.

Mostly household in Pakistan pay for the waste collection from doorstep. This exactly can be seen that if van/cart collect the waste from door step, households' cash payment for waste collection and disposal increases both positively and significantly. This is true in case of Pakistan, urban, and rural areas, respectively. Similarly, van/cart services positively and significantly impact the households' cash payment for waste disposal in Punjab, Sindh, and Baluchistan provinces, respectively (Table 3). However, in Khyber Pakhtunkhwa van/cart services for waste collection significantly and negatively impact the cash payment. It could be because the households in Khyber Pakhtunkhwa may use other services for waste collection rather than cart/van services. In the other studies, Endalew & Tassie [4] and Tassie & Endalew [33] linked the waste collection from doorstep with households' willingness to pay for solid waste management services. They found that that access to solid waste management services positively affect the households' willingness to pay for solid waste management services.

Infrastructure availability is most important factor affecting the households' cash payment for waste disposal. If infrastructure availability is present, it can reduce households' efforts and cash. For instance, if the availability of bin or dumpster or waste collection point is there in the nearby locality, households' cash payment significantly reduces in Pakistan (Table 2). Same is true for urban area where cash payment reduces, however, infrastructure availability such as bins or dumpsters or waste collection point are not enough to capture any meaningful impact of it on rural households' cash

payment of waste management. Table 3 depicts that bin/dumpster/waste collection point availability negatively impact the cash payment in Sindh and Khyber Pakhtunkhwa. It means that access to public bin/dumpster/collection point led households to pay less cash for waste disposal. This may be the reason people throw waste in/around the streets, open pits, ponds, rivers, and agricultural land and that is why waste management condition is worst in these provinces (Pak-EPA, 2024). However, bin/dumpster/waste collection point availability positively affect the cash payment in Baluchistan. It means that access to public bin/dumpster/collection point led households to pay more cash for waste disposal in Baluchistan. In other studies, Chukwuone et al. [20] linked the availability of dumpster with illegal waste disposal and found that the availability of dumpster in a residential site significantly reduce the likelihood of illegal waste disposal.

Average time the household spent on a round trip to the nearest public bin/collection point significantly affect the households cash payment for waste collection and disposal services in Pakistan and urban areas. However, when the average time to the nearest public bin increases the households cash payment increase reached to minimum (11–15 minutes for Pakistan and 16–20 minutes for urban areas) then increasing again when the average time to the nearest public bin increases further. This shows that if the waste disposal facility sites are situated near to the residential sites such that on the distance of 1–11 minutes for Pakistan and 1–15 minutes for urban areas, the households cash payment for waste collection and disposal services increase and fall when reaches to a certain threshold. However, beyond a certain threshold if the distance from solid waste disposal facility increases the households cash payment increases again. This is because if the longer the distance the more complicate problem of waste collection and disposal as households would have to walk a long distance to dispose their waste. In Sindh, when the average time to the nearest public bin/collection point increases from 1–5 minutes to 6–10 minutes the households cash payment for waste collection and disposal services increases. Conversely, when the distance increases from 11–15 minutes to 21–25 minutes the households cash payment for waste collection and disposal services decreases. In Khyber Pakhtunkhwa, when the average time to the nearest public bin or collection point is 1–5 minutes or 11–15 minutes, household cash payments for waste collection and disposal services increase. However, the cash payment decreases at a distance of 6–10 minutes. Other studies linked the distance between resident area and waste disposal facility with households' willingness to pay for improved solid waste management services and found a significant positive effect of this variable on the household willingness to pay for improved solid waste management services [36,40].

In Pakistan, if the nearest public bin/collection point available to households is emptied/cleared by the concern person(s) or local authority(s) twice or thrice a week then households are pay more cash for waste collection and disposal services. Same results are obtained for urban areas of the country. This shows that if the bins/collection points are emptied by the concern person(s) or local authority(s) after few days then households cash payment for waste disposal increases. It is reasonable because, if the bin/collection point is not emptied on time than it can cause various diseases and can adversely affect the households' health. In order to avoid the health problem, the households whose bins/collection point are cleared after few days such that twice or thrice a week could pay more cash for waste disposal services. On the other hand, in rural areas of the country, if the nearest public bin/collection point available to households is emptied/cleared by the concern person(s) or local authority(s) once or twice a week then households are pay less cash for waste collection and disposal services. The duration of the public bin/collection point clearance is not the matter of rural households because most of the rural households may prefer to throw their waste (mostly animal dung) in their agricultural fields rather than to dispose it in public bin/collection point. In Punjab, if the bin/collection point is cleared twice or thrice a week then households are willing to pay more cash for waste collection and disposal services while if the bin/collection point is cleared once a week then households are willing to pay less cash for waste collection and disposal services. In Sindh, if the nearest public bin/collection point is emptied every day then households are pay more cash for waste collection and disposal. However, in Baluchistan, the households cash payment for waste collection and disposal services decreases regardless of the duration of the bin/collection point cleared. Furthermore, in Khyber Pakhtunkhwa, the households cash payment for waste collection and disposal services increases if the bin/collection point cleared once a week. This shows

that households in Punjab, Sindh, and Khyber Pakhtunkhwa may be more conscious about the adverse health impact of solid waste as compared to the households in Baluchistan.

If households own the house, they pay more cash for waste collection and disposal services. This is both true in case of Pakistan and urban areas (Table 2). However, in rural areas, ownership of house did not provide enough evidence to capture any meaningful expression with households' cash payment for waste disposal. However, if number of rooms in a rural household house increase, then they pay less cash for waste collection and disposal. Although larger houses accommodate more people, the per-room waste generation may be lower compared to smaller and more densely occupied houses. Province-wise breakdown reveals that only in Punjab, there is positive and significant impact of ownership of house on cash payment (Table 3). It means that if households own a house, they pay more cash for waste collection and disposal. Results are insignificant on all other provinces. In other studies, Adzawla et al. [35] linked the housing characteristics with households' decision to adopt a particular solid waste disposal system and found a negative impact of own housing on households' decision to adopt a particular solid waste disposal system. Similarly, Ahmed [40] linked house ownership with households' willingness to pay for improved solid waste management services and found a positive impact of house ownership on the willingness to pay for improved solid waste management services.

Dwelling type of households negatively but significantly affect the households cash payment for waste disposal and collection. For instance, if households live independently (unlike in joint family), they pay less cash for waste collection and disposal (Table 2). Households who live independently they carry on all responsibilities including the waste management. They can easily dispose-off waste on approved places and can save cash. This is true in both Pakistan and urban areas. Similar results can be seen for dwelling type such as households who lives in apartments, part of large unit and compound. Almost similar results can be seen for various type of dwelling in Punjab and Sindh. For instance, dwellers of apartment and part of compound pay less cash for waste collection and disposal in Punjab while dwellers of independent house, part of large unit, and part of compound pay less cash for waste collection and disposal in Sindh (Table 3). Usually management of apartments, large units (of residential areas) and compounds manage all the wastes. In such cases, residents of such dwelling type do not need to pay cash for waste disposal. In other studies, Adzawla et al. [35]linked the dwelling type with households' decision to adopt a particular solid waste disposal system and found a positive impact of compound housing on households' decision to adopt a particular solid waste disposal system.

Urban households in Pakistan, Punjab, and Khyber Pakhtunkhwa pays significantly less cash payment for waste disposal as compared to counterfactual (e.g., rural households). In urban areas of the country, especially in the urban areas of Punjab and Khyber Pakhtunkhwa a well-developed system of waste collection and disposal is available to the households. This system properly works under the supervision of waste management companies and water and sensation agencies. Therefore, the urban households pay less cash for waste disposal as compared to rural households. However, in Baluchistan the opposite results are true. In Baluchistan, urban residents pay significantly more cash payment for waste disposal as compared to rural households. The system of waste collection and disposal available to households in Baluchistan is not organized nor efficient as compared to Punjab and Khyber Pakhtunkhwa. Therefore, the urban households in Baluchistan pay more cash for waste disposal as compared to rural households. In other studies, Adzawla et al. [35] linked the regional dummies with households' decision to adopt a particular solid waste disposal system and found ambiguous impact of regional dummies on households' decision to adopt a particular solid waste disposal system.

The study also compared the cash payment of different provinces to better understand the picture. For instance, households in Punjab, Sindh, and Baluchistan significantly pay less cash for waste disposal services as compared to reference category, Khyber Pakhtunkhwa (Table 2). Similarly, urban and rural households in Punjab, Sindh, and Baluchistan pay less cash for waste disposal services as compared to urban and rural households in Khyber Pakhtunkhwa. As compared to other provinces the Khyber Pakhtunkhwa is rich in natural beauty and thus have abundant natural recreation sites and clean cities such that Peshawar is called the city of flowers and district Swat is called the Switzerland of Asia. For keeping

the clean and neat environment of the beautiful natural sites and cities the households in this province pay more cash for waste collection and disposal services as compared to the households in other provinces.

## Conclusion and recommendations

This study aims to evaluate the existing solid waste collection and management system available to households in Pakistan. The study also explores the factors affecting the households cash payments for waste collection and disposal services. The robust least squares regression results show that income negatively affects households' cash payments for solid waste collection and disposal services in Pakistan, urban areas, and Sindh but income has no significant impact in rural areas, Khyber Pakhtunkhwa, and Baluchistan. Households with educated or older heads and female-headed households pay more for solid waste collection and disposal services in Pakistan, urban areas, and Sindh. Moreover, an increase in the number of income earners reduce payments for solid waste collection and disposal services in Pakistan, Baluchistan, and Khyber Pakhtunkhwa. Doorstep waste collection increases payments for solid waste collection and disposal services in Pakistan, urban and rural areas, Punjab, Sindh, and Baluchistan, while availability of nearby bins/collection points lower the payments in Pakistan, urban areas, Sindh, and Khyber Pakhtunkhwa. Nearness to waste disposal sites raises payments for solid waste collection and disposal services in Pakistan, urban areas, Sindh, and Khyber Pakhtunkhwa. Households in Pakistan, urban areas, and Punjab pay more for solid waste collection and disposal services when public bins are emptied twice or thrice weekly. House ownership also pay more in Pakistan, urban areas, and Punjab. Finally, house ownership is associated with higher cash payments for waste collection and disposal services, a trend observed in Pakistan, urban areas, and Punjab.

Based on the above results we come to an end that an increase in household income can bring significant increase in their cash payment for solid waste collection and disposal services. Thus, efforts to increase the households' incomes at national and provincial level. The findings related to socio-demographic factors, such as the education and age of the household head and female-headed households, are also significant. Based on these results, it is recommended that federal, provincial, and local governments in Pakistan provide women with greater decision-making opportunities in waste management to achieve improved outcomes. Furthermore, these findings highlight the need for proper education and awareness programs among Pakistani households on sustainable waste disposal practices. These programs will enable the households to pay more for the improved waste collection and disposal services. While all households require education on waste management, female-headed households should be given priority. Furthermore, relevant waste collection and disposal authorities, including waste management companies, municipal committees, local government bodies, and policymakers, should take socio-demographic factors into account when designing their waste collection and disposal services policies and determining service charges.

The positive impact of waste collection via van/cart and households' cash payment for waste collection and disposal services is an impetus for municipal corporations, waste collection agencies, water and sanitation corporations and other private waste collection companies to further strengthen the van/cart facilities that the households dispose their waste at door and hence in return they will get batter cash for this service. However, we found opposite story in Khyber Pakhtunkhwa where the same facility at doorstep reduces the households cash payments. Therefore, alternative waste management policies are needed for these households which enable them to convert their waste to organic fertilizers. Besides, the results indicates that public bin or waste collection point availability is not enough to solve the waste management problems in the country and to increase the cash payment for waste collection. The real problem of waste management in the country is the distance from the waste disposal facility and the duration of clearness of waste disposal facility. Therefore, it is recommended that the authorities put the bin or collection point near to the households' residence sites that they are dispose their waste in lesser time and lesser efforts. The second problem of waste management, is these bins or collection points are emptied/cleared by the concern person(s) or local authority(s) after few days. If the bin or collection point is not emptied on time than it can cause various diseases and can adversely affect the households' health.

Therefore, it is recommended that the authorities must devise proper and quick method for clearing the bins or collection points every day. Finally, the house ownership can significantly increase the households cash payment for waste collection and disposal services.

## Supporting information

**S1 Data. Data Set of all variables.**
(XLSX)

## Acknowledgments

Authors acknowledge colleagues of Department of Agricultural Extension and Rural Society, King Saud University for providing their invaluable guidance, feedback, and support throughout our research..

## Author contributions

**Conceptualization:** Ghulam Mustafa.

**Data curation:** Naveed Hayat, Bader Alhafi Alotaibi.

**Formal analysis:** Ghulam Mustafa, Naveed Hayat, Abou Traore.

**Funding acquisition:** Bader Alhafi Alotaibi, Abou Traore.

**Investigation:** Ghulam Mustafa, Naveed Hayat, Bader Alhafi Alotaibi, Abou Traore.

**Methodology:** Ghulam Mustafa, Naveed Hayat, Abou Traore.

**Project administration:** Naveed Hayat, Bader Alhafi Alotaibi, Abou Traore.

**Resources:** Bader Alhafi Alotaibi.

**Software:** Bader Alhafi Alotaibi, Abou Traore.

**Supervision:** Ghulam Mustafa.

**Validation:** Abou Traore.

**Visualization:** Naveed Hayat, Bader Alhafi Alotaibi.

**Writing – original draft:** Ghulam Mustafa.

**Writing – review & editing:** Naveed Hayat, Bader Alhafi Alotaibi, Abou Traore.

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
