## [Decision Letter · Decision Letter 0]

Dear Dr. Mustafa,

Thank you for submitting your manuscript to PLOS ONE. After careful consideration, we feel that it has merit but does not fully meet PLOS ONE’s publication criteria as it currently stands. Therefore, we invite you to submit a revised version of the manuscript that addresses the points raised during the review process.

In addition to the criticisms below, this typo also needs to be corrected.  (Bennet**<del>s</del>**

We look forward to receiving your revised manuscript.

Kind regards,

Hatime Kamilcelebi

Academic Editor

PLOS ONE

Journal Requirements:

 This research was funded by Researchers Supporting Project Number (RSP2024R443), King Saud University, Riyadh, Saudi Arabia.

Reviewers' comments:

Reviewer's Responses to Questions

**Comments to the Author**

1. Is the manuscript technically sound, and do the data support the conclusions?

Reviewer #1: Yes

Reviewer #2: Partly

2. Has the statistical analysis been performed appropriately and rigorously?

Reviewer #1: Yes

Reviewer #2: No

3. Have the authors made all data underlying the findings in their manuscript fully available?

Reviewer #1: Yes

Reviewer #2: Yes

4. Is the manuscript presented in an intelligible fashion and written in standard English?

Reviewer #1: Yes

Reviewer #2: No

Reviewer #1: The manuscript offers valuable insights into Pakistan's waste management system, and the statistical analysis is both thorough and satisfying. However, I have highlighted some areas in the attached document where corrections are necessary. Firstly, the general statistics provided in the introduction need clearer and more precise references. Additionally, some figures should be redesigned to enhance clarity and visual impact.

Reviewer #2: Reviewer Comments

Summary

This paper addresses an important environmental issue of our time- solid waste management in a developing country context. It sought to contribute to knowledge on the factors that determine household payment for solid waste management. The authors cited related studies from different countries. The authors used secondary national data for the study. The findings from this study will contribute to policy and dialogue on solid waste management in Pakistan and similar developing countries. However the paper needs to clearly establish the gap in knowledge it seeks to fill and state the objectives more clearly. It also needs to be better organized in terms of methodology, presentation and grammar.

Abstract

The abstract is a good summary of paper.

Introduction

The authors need to describe a little bit more of the existing waste management system in Pakistan. Is it public sector led or private sector led? Is there synergy between public sector and public sector in waste management? What policy directions does the government currently pursue for waste management. That will give the reader a better context for the paper. The objective should be stated more explicitly. To ‘document’ waste system as an objective, should be restated.

Manuscript Formatting

The next section after introduction is labelled “Manuscript Formatting’ and this appears like a misnomer. The authors should crosscheck if that is what the authors really want to name that section.? It seems materials and methods or methodology should be a better option.

The study Area should be a sub- section in this Materials and Methods or Methodology section. Here the authors should describe the area of study briefly. How many provinces in all are in Pakistan? Was the data used across the whole of Pakistan? If the data is a nationally representative data, then the authors should discuss Pakistan socio-demographics relevant to their study. What are the sources of livelihood in the different provinces, especially the rural areas that earn more income than urban areas. Also a map of the country where the major provinces highlighted in the results section is distinctively shown would be appropriate.

Data

This section needs to describe the data set a bit more. How were the samples chosen by the Bureau of statistics?. Is this the only wave available for Pakistan ? Why did the authors not use panel data if there are other waves?.

Empirical Methods

This section presented by the authors seems to be the analytical framework. What about the empirical strategy? The authors should specify the model – robust least squares regression they used. What is your identification strategy for establishing causality? The authors should cross check what gi is in the various equations where it is specified and make sure there are no errors. It assumes different variables in the two equations where it is specified.

The authors should reconsider this empirical method section in terms of model used……. There are a few socio-demographic variables expected to be included like education of household head, gender of household head, number of income earners in household which the authors did not include and did not provide reasons for non-inclusion.

Results and Findings

The first figure is not part of your findings. It should be moved to the data section, under sampling. Figure labels are usually placed after the figure not before the figure. This style runs through all the figures the authors presented and should be corrected. Figure 2 label should go below the figure. It appears it is mean cash payment. If so then it should be reflected in the figure label.

The mean household payment for waste collection should come after Mean household income. Did the authors try to explore why monthly mean income is higher in rural areas than urban areas from literature. It would be good to state it here, because the reverse is expected. Also the authors should explore why payment for waste is higher in rural areas from literature. Apriori we expect rural areas to be agriculture based, and to use decomposable waste for home gardens, therefore minimizing waste quantity to be discarded, and associated cost. The authors should try to find from more microlevel waste study in Pakistan why there is higher payment in rural areas aside from the reasons they gave.

For the descriptive statistics, it would be good to include the minimum and maximum values of the variables.

Was the household waste management techniques mutually exclusive? Like a household using public waste collector did not at the same time use private? A household using the above two did not have a bin accessible to them?

3.3 Factors affecting cash payments

The authors need to be careful in drawing inferences from the data. For example the authors state that ‘ However, in Punjab (Table 3) we observed that an increase in household income increase their cash payment for solid waste collection and disposal services whereas in Sindh an increase in household income brings a significant decrease in their cash payments for waste collection and disposal services. Punjab is the largest province of Pakistan by population, the province has relatively developed agriculture, industrial and services sector as compared to other provinces. Therefore, the households in this province give proper attention to waste management and hence their cash payment for waste collection and disposal increases as their income increases. Conversely, by population Sindh is the second largest province of the country, the province is poor as compared to the Punjab Therefore, any increase in the households’ income decrease their payment on waste disposal’ see lines 286-294. Instead state that ‘maybe due to……., or could be because…….’ Cross check every where in this section for similar inferences.

Line 341 contradicts what you stated in earlier section that rural households are richer see line 186-187.

The authors need to take another look at the interpretation of their data. It goes back to the empirical strategy given the nature of their data. There are many categorical explanatory variables in the data. They should explore a more suitable model for analysis.

Conclusion and recommendation

The authors essentially repeated almost verbatim what they stated in the results section. The conclusion should be a brief summary of the study highlighting study objectives, empirical approach and major findings then the recommendations which should inform policy direction.

References

Some cited papers are not in reference list. Example – Kaza and Lisa 2018; Tadesse et al 2007. The authors should consider providing the DOI of the papers. 

Major issues

The authors should state the empirical model used, review if that empirical model best suits the data available. They should also cross check the interpretation of the results.

Minor Issues

The authors should consider enlisting the services of a writing coach to improve the grammar and flow of the write-up.

**Do you want your identity to be public for this peer review?** For information about this choice, including consent withdrawal, please see our Privacy Policy

Reviewer #1: **Yes: ** Billur Engin Balin

Reviewer #2: No

---

## [Author Response · Author response to Decision Letter 1]

8 Nov 2024

Editorial Team Comment: In addition to the criticisms below, this typo also needs to be corrected. (Bennetst, 2008) at the line 72.

Response to Editorial team: We address this typo.

Editorial Team Comment: Please include the following items when submitting your revised manuscript:

Response to Editorial team: There is rebuttal from authors. All comments are addressed.

Editorial Team Comment: A marked-up copy of your manuscript that highlights changes made to the original version. You should upload this as a separate file labeled 'Revised Manuscript with Track Changes'.

Response to Editorial team: Marked-up copy is provided that highlighted changes.

Editorial Team Comment: An unmarked version of your revised paper without tracked changes. You should upload this as a separate file labeled 'Manuscript'.

Response to Editorial team: It is provided

Editorial Team Comment: When submitting your revision, we need you to address these additional requirements.

Response to Editorial team: It is provided

Reviewers' comments:

Reviewer's Responses to Questions

Comments to the Author

1. Is the manuscript technically sound, and do the data support the conclusions?

Reviewer #1: Yes

Reviewer #2: Partly

Response to Reviewers: We re-regress the model, clarified the data and wrote findings and hence conclusion based on these.

2. Has the statistical analysis been performed appropriately and rigorously?

Reviewer #1: Yes

Reviewer #2: No

Response to Reviewers: We re-regress the model and provided its justifications.

3. Have the authors made all data underlying the findings in their manuscript fully available?

Reviewer #1: Yes

Reviewer #2: Yes

Response to Reviewers: Thank you for appreciating.

4. Is the manuscript presented in an intelligible fashion and written in standard English?

Reviewer #1: Yes

Reviewer #2: No

Response to Reviewers: We re-write the paper. Paper quality is improved based on your comments. Please let us know how we can improve it further.

5. Review Comments to the Author

Reviewer #1:

Reviewer #1: The manuscript offers valuable insights into Pakistan's waste management system, and the statistical analysis is both thorough and satisfying. However, I have highlighted some areas in the attached document where corrections are necessary. Firstly, the general statistics provided in the introduction need clearer and more precise references. Additionally, some figures should be redesigned to enhance clarity and visual impact.

Response to Reviewer: Thank you very much for reviewing our paper. We addressed all comments and there is no rebuttal. We cited all the statistics and revised the Figures under your valuable comments.

Reviewer Comment: This beginning is a kind of standard beginning for an introduction. Do we really need to refer to another paper?

Response to Review: We revised the introduction based on your comment. In opening of this paragraph two (Line 46-60). We changed it based on your and other reviewers’ comments. The study highlights some factors that cause solid waste. Therefore, we think these claims should be properly cited. Moreover, we added new opening of introduction part. Please see Lines 33-44.

Reviewer Comment: The rest of the paragraph provides statistics on the topic. However, based on the referencing, it may be misunderstood that the authors are the source of these statistics. For instance, it is mentioned that the annual global generation of solid waste exceeds 2 billion tonnes; however, this statistic actually comes from a World Bank study. Therefore, it would be more accurate to emphasize that the data is sourced from the World Bank within the text.

Response to Reviewer: We incorporated this change. See line 46-57

Reviewer Comment: "I recommend adding a paragraph that outlines the structure of the study. For example:

The remainder of the paper is organized as follows: ...."

Response to Reviewer: We incorporated this change. See last paragraph of introduction from line 148 to 151.

Reviewer Comment: This figure is somewhat confusing. I suggest dividing it into two separate figures. The first (figure 1.a.) could display Pakistan – Urban – Rural (perhaps using a pie chart). The second figure (figure 1.b.) could focus on the provinces, which could be represented using either a bar chart or another pie chart.

Response to Reviewer: We incorporated this change. Please see Figs 2, 3, 4 and 5.

Reviewer comment: "Once again, I believe the figure should be divided. If it’s not divided, then the colors used in the figures must be designed consistently for clarity."

Response to Reviewer: We incorporated this change. Please see Figs 2, 3, 4 and 5.

Reviewer comment: Same comment like previous two comment.

Response to Reviewer: We incorporated this change. Please see Figs 2, 3, 4 and 5.

Reviewer #2: Reviewer Comments

Summary

This paper addresses an important environmental issue of our time- solid waste management in a developing country context. It sought to contribute to knowledge on the factors that determine household payment for solid waste management. The authors cited related studies from different countries. The authors used secondary national data for the study. The findings from this study will contribute to policy and dialogue on solid waste management in Pakistan and similar developing countries. However the paper needs to clearly establish the gap in knowledge it seeks to fill and state the objectives more clearly. It also needs to be better organized in terms of methodology, presentation and grammar.

Response to Reviewer: Thank you very much for reviewing our paper. We really learnt from your comments. We are happy to incorporate those changes and there is no point of rebuttal. We tried to establish gap in the literature (See highlighted parts of introduction) and we also clarified our objectives of the study (See line 143-146). We improved our methodology based on your comments (See methodology part) and presented findings of our study based upon the information gathered as a result of the methodology (See Results and Discussion).

Abstract

Reviewer Comment: The abstract is a good summary of paper.

Authors Response: Thank you for appreciating the Abstract

Reviewer Comment:

Introduction

The authors need to describe a little bit more of the existing waste management system in Pakistan. Is it public sector led or private sector led? Is there synergy between public sector and public sector in waste management? What policy directions does the government currently pursue for waste management. That will give the reader a better context for the paper. The objective should be stated more explicitly. To ‘document’ waste system as an objective, should be restated.

Response to Reviewer: We address this comment and revised our introduction based on your comment. Solid waste collection is owned and operated by government of Pakistan. However, in some metropolitan cities public-private partnership is there. We provide the complete synergy between public and private sector. Revised introduction can be seen in first paragraph of introduction (Lines 33-44). Waste management system in Pakistan and its four province along with public-private partnership can be seen in Paragraph 6, 7, 8 and 9 of Introduction (Lines 94-131). The objectives are restated and clarified (Please See line 143-146).

Reviewer Comment:

Manuscript Formatting

The next section after introduction is labelled “Manuscript Formatting’ and this appears like a misnomer. The authors should crosscheck if that is what the authors really want to name that section.? It seems materials and methods or methodology should be a better option.

Response to Reviewer: Thank you very much for highlighting typo. We changed it into Materials and methods

Reviewer Comment: The study Area should be a sub- section in this Materials and Methods or Methodology section. Here the authors should describe the area of study briefly. How many provinces in all are in Pakistan? Was the data used across the whole of Pakistan? If the data is a nationally representative data, then the authors should discuss Pakistan socio-demographics relevant to their study. What are the sources of livelihood in the different provinces, especially the rural areas that earn more income than urban areas. Also a map of the country where the major provinces highlighted in the results section is distinctively shown would be appropriate.

Response to Reviewer: We address these comments by following ways;

a. The study Area is provided under Materials and Methods section. We also described the area of study. See Lines 154to 192.

b. We briefly described Pakistan and its four provinces. The data used in the study represent the whole Pakistan. The data is nationally represented that is collected through House-hold Integrated Economic Survey (HIES) 2018-2019. We briefly described the socio-economic status of Pakistan and its four provinces under the heading “Study Site”. We also provided livelihood status of different provinces (See Paragraph 2 and 3 under heading “Study Site”). See Lines 154 to 192

c. Map of the study showing different province is also provided. Please see Fig 1.

Reviewer Comment:

Data

This section needs to describe the data set a bit more. How were the samples chosen by the Bureau of statistics?. Is this the only wave available for Pakistan ? Why did the authors not use panel data if there are other waves?.

Response to Reviewer: We address this comment in the first paragraph under the heading “Data” and briefly justify here. This study used the recent data of the Pakistan Social Living Measurement Survey (PSLM) for the year 2019-2020, conducted by the Pakistan Bureau of Statistics. PSLM, 2019-20 is the twelfth round of a series of surveys, initiated in 2004. The first round of the survey started in 2004-2005 and up to date the recent most round survey data is available for the year 2019-20. However, the survey is not carried out for two round such that for 2009-10 and for 2017-18. Besides in each round the sample size is changed. Thus, making a panel data from various round of PSLM is not possible. Thus, we use the most recent data of the 12th round of PSLM for the year 2019-2020, which include 160,654 households throughout Pakistan. However, some households did not report the information on their cash payments for waste collection and disposal services, so after organizing the data, we used data from 16,155 households who pay cash for waste collection and disposal services. For further detail, please see Lines 197-213.

Reviewer Comment:

Empirical Methods

This section presented by the authors seems to be the analytical framework. What about the empirical strategy? The authors should specify the model – robust least squares regression they used. What is your identification strategy for establishing causality? The authors should cross check what gi is in the various equations where it is specified and make sure there are no errors. It assumes different variables in the two equations where it is specified.

Response to Reviewer: We added sub-heading under Material and Methods namely; Analytical Framework and Estimation strategy in the light of your comment. We incorporate these under heading of Analytical Framework. Please see Lines 224-227 and 236-237.

Reviewer Comment: The authors should reconsider this empirical method section in terms of model used……. There are a few socio-demographic variables expected to be included like education of household head, gender of household head, number of income earners in household which the authors did not include and did not provide reasons for non-inclusion.

Response to Reviewer: We address this comment under the Sub-subheading of empirical estimation strategy. Please paragraph 2, 3 and 4 (See line 250-253, 263-296) under Empirical Estimation Strategy. We also address your comment regarding education, gender, number of income earners in the last paragraph of Conclusion.

Results and Findings

Reviewer Comment: The first figure is not part of your findings. It should be moved to the data section, under sampling. Figure labels are usually placed after the figure not before the figure. This style runs through all the figures the authors presented and should be corrected. Figure 2 label should go below the figure. It appears it is mean cash payment. If so then it should be reflected in the figure label.

Response to Reviewer: We moved this Figure in Material and Methods section Please see Figure 1 and 2. Similar comment given by other reviewer so we break this Figures into two figures. Moreover, we used journal style for figures and tables.

Reviewer Comment: The mean household payment for waste collection should come after Mean household income. Did the authors try to explore why monthly mean income is higher in rural areas than urban areas from literature. It would be good to state it here, because the reverse is expected. Also the authors should explore why payment for waste is higher in rural areas from literature. Apriori we expect rural areas to be agriculture based, and to use decomposable waste for home gardens, therefore minimizing waste quantity to be discarded, and associated cost. The authors should try to find from more microlevel waste study in Pakistan why there is higher payment in rural areas aside from the reasons they gave.

Response to Reviewer: Mean cash payment is added. Please see Figure 6. Moreover, we address other part of comment in first paragraph of Results and Discussion under the subheading “Existing solid waste collection…”. See line 306-314.

Reviewer Comment: For the descriptive statistics, it would be good to include the minimum and maximum values of the variables.

Response to Reviewer: We included minimum and maximum values of the variables. Please see Table 1.

Reviewer Comment: Was the household waste management techniques mutually exclusive? Like a household using public waste collector did not at the same time use private? A household using the above two did not have a bin accessible to them?

Response to Reviewer: We incorporated this comment from Line 366 to Line 376.

3.3 Factors affecting cash payments

The authors need to be careful in drawing inferences from the data. For example the authors state that ‘ However, in Punjab (Table 3) we observed that an increase in household income increase their cash payment for solid waste collection and disposal services whereas in Sindh an increase in household income brings a significant decrease in their cash payments for waste collection and disposal services. Punjab is the largest province of Pakistan by population, the province has relatively developed agriculture, industrial and services sector as compared to other provinces. Therefore, the households in this province give proper attention to waste mana

---

## [Decision Letter · Decision Letter 1]

Dear Dr. Mustafa,

Thank you for submitting your manuscript to PLOS ONE. After careful consideration, we feel that it has merit but does not fully meet PLOS ONE’s publication criteria as it currently stands. Therefore, we invite you to submit a revised version of the manuscript that addresses the points raised during the review process.

We look forward to receiving your revised manuscript.

Kind regards,

Hatime Kamilcelebi

Academic Editor

PLOS ONE

Journal Requirements:

Reviewers' comments:

Reviewer's Responses to Questions

**Comments to the Author**

Reviewer #1: All comments have been addressed

Reviewer #2: (No Response)

2. Is the manuscript technically sound, and do the data support the conclusions?

Reviewer #1: Yes

Reviewer #2: Partly

3. Has the statistical analysis been performed appropriately and rigorously?

Reviewer #1: Yes

Reviewer #2: Yes

4. Have the authors made all data underlying the findings in their manuscript fully available?

Reviewer #1: Yes

Reviewer #2: Yes

5. Is the manuscript presented in an intelligible fashion and written in standard English?

Reviewer #1: Yes

Reviewer #2: No

Reviewer #1: Dear Authors,

I have reviewed the revised manuscript and observed that all the requested revisions have been addressed thoroughly. Thank you for your effort and attention to detail. I recommend the manuscript for publication.

Best regards,

Reviewer #2: General Comments:

The authors have revised the work based on earlier comments, but a few comments were not effected, like adding some more variables in the regression that was suggested.They added it in conclusion as suggestion for further study. In revising the work, the authors paid little attention to the grammar and sentence structure. A few examples of corrections that need to be done follow:

Line 36, remove Now a days. Start the sentence with ‘Developed and developing countries……

Line 46: ….. the demand for solid waste management services has increased. ( insert ‘services’ after waste management)

Line 49:…… billion in 2025; with this, the waste generation….. ( insert ‘with’ before this the waste management…).

Line 56: in developing countries…….. 50 percent of municipal operation budget is spent on… 9

(insert ‘is’ before spent on…)

Line 78- 79

Line 83-85: users of solid waste management services pay higher amount than ???? for waste management ( something is missing there. If you cant supply that information, then remove that last sentence.

Line 86: usually people are reluctant to pay for waste disposal….(insert ‘are’)

Line 136: Infrastructure which create serious environmental….. ( replace creating with create)

Line 138: The country is facing enormous challenges …( insert is after country)

Line 145: Findings of this study will guide the policy makers in …( remve provide help)

Line 148:

Line 152. Please choose Methodology instead of materials and Methods. ( materials and methdos are usually for laboratory based research work.

Line 156: in the country, 61 percent …population live in rural areas ( use live instead of living, make same changes in line 157)

Line 159 Remove the statements after the Arabian Sea and Golf of Oman in the south. End the sentence here and delete the rest of information on countries bothering Pakistan.

Further summarize the study site please.

Line 199-201: recast as follows: The first round of the survey started in 2004-2005 and upto date the most recent round of survey data was conducted in year 2019-20. The survey was not carried out for 2 rounds -2009-10 and for 2017-18. ( Please change your sentence with this suggested).

LLine 300: form fig 4 we… ( change Form to from)

Line 306: …first, Rural incomes ( remove capital ;R” in Rural and use small ‘r’. cross check other sentences for such cases in the entire work.

Line 338: the results can have two compensations ( remove compensation and replace with ‘possible interpretations’)

Line 341; …. This might be the reason that there is well established ( I suggest ….’It could be because there is well established…).

Line 345: Secondly, waste management is a need rather than choice… ( I suggest: waste management service is important, so people spend money for it…)

Line 347-348 ( is Akhter’s finding in relation to a month or a year? It is good to state whether it is USD 4.8 per month or per year.)

Line 370 -373 Please cross check the information in these sentences with what you have in the figure. In the discussion you have 83% with waste collection bin and 17% without waste collection bill, but your figure is saying the opposite. There is inconsistency. In figure 9, you have bin/ waste collection point not available/accessible 83% but the arrow is pointing to the small portion of the pie. You have bin/waste collection point is available /accessible by 13% but the arrow is pointing to larger portion of the pie.

Line 394 -397 : is it possible for you to find the mean/ average time for all respondents to get to the bin/collection point? That will help you if really most spend a lot of time disposing waste. From literature, is there a recommended time to be spent disposing waste? Beyond which time it is regarded as too much time? If there is such information and you can cite it, it will make your discussion more interesting.

Line 439-617: please carefully make sentence corrections to convey intended meanings to readers.

Line 437: Factors affecting the households cash payments for waste collection and

disposal services

The regression reported here is only the robust least squares. Is it possible to report the OLS that was tried before testing for heteroskedasticity. It can be reported in same tables, side by side with the robust least squares. It will help readers to appreciate what you stated about the OLS and the robust least squares.

Please check out your sentences especially in the discussion of the regression results.

**Do you want your identity to be public for this peer review?** For information about this choice, including consent withdrawal, please see our Privacy Policy

Reviewer #1: No

Reviewer #2: No

---

## [Author Response · Author response to Decision Letter 2]

10 Mar 2025

Reviewer #1: Dear Authors,

I have reviewed the revised manuscript and observed that all the requested revisions have been addressed thoroughly. Thank you for your effort and attention to detail. I recommend the manuscript for publication.

Response: Dear reviewer thank you for your appreciation.

Reviewer #2: General Comments:

1. The authors have revised the work based on earlier comments, but a few comments were not effected, like adding some more variables in the regression that was suggested. They added it in conclusion as suggestion for further study.

Response: Special thanks for this comment, as per your suggestion, we included socio-demographic variables like education of the household head, age of the household head, gender of household head, and number of income earners in the household. The inclusion of these variables furthers improve the regression results (for details see lines 259-262; 420-424; 448-466; 472-489; 493-543; 546-555; 564-569; 572-597; 603-605; 609-611; 614-617; 621-624; 639-642; 653-658 Table 1; Table 2; and Table 3). We also revised the conclusion and recommendation part based on new results.

2. In revising the work, the authors paid little attention to the grammar and sentence structure. A few examples of corrections that need to be done follow:

Line 36, remove Now a days. Start the sentence with ‘Developed and developing countries……

Response: Incorporated. Please see Line# 37

Line 46: ….. the demand for solid waste management services has increased. ( insert ‘services’ after waste management)

Response: Incorporated. Please see Line# 47

Line 49:…… billion in 2025; with this, the waste generation….. ( insert ‘with’ before this the waste management…).

Response: Incorporated. Please see Line# 50

Line 56: in developing countries…….. 50 percent of municipal operation budget is spent on… 9

(insert ‘is’ before spent on…)

Response: Incorporated. Please see Line# 57

Line 78- 79

Response: Please let us know what is issue in these lines?

Line 83-85: users of solid waste management services pay higher amount than ???? for waste management ( something is missing there. If you cant supply that information, then remove that last sentence.

Response: We removed this sentence.

Line 86: usually people are reluctant to pay for waste disposal….(insert ‘are’)

Response: Incorporated. Please see Line# 84

Line 136: Infrastructure which create serious environmental….. (replace creating with create)

Response: Incorporated. Please see Line# 135

Line 138: The country is facing enormous challenges …( insert is after country)

Response: Incorporated. Please see Line# 137

Line 145: Findings of this study will guide the policy makers in …( remve provide help)

Response: Incorporated. Please see Line# 144

Line 148:

Response: Please let us know what is issue in this line?

Line 152. Please choose Methodology instead of materials and Methods. ( materials and methdos are usually for laboratory based research work.

Response: Thank you very much for this comment. It is the learning of a day for me. Incorporated. Please see Line# 151

Line 156: in the country, 61 percent …population live in rural areas ( use live instead of living, make same changes in line 157)

Response: Incorporated. Please see Line# 155 and 156

Line 159 Remove the statements after the Arabian Sea and Golf of Oman in the south. End the sentence here and delete the rest of information on countries bothering Pakistan.

Further summarize the study site please.

Response: We incorporated these changes and summarized this part. Please see lines#153-159.

Line 199-201: recast as follows: The first round of the survey started in 2004-2005 and upto date the most recent round of survey data was conducted in year 2019-20. The survey was not carried out for 2 rounds -2009-10 and for 2017-18. ( Please change your sentence with this suggested).

Response: Incorporated. Please see Lines # 200-202. Thank you for very nice suggestion.

Line 300: form fig 4 we… ( change Form to from)

Response: Incorporated. Please see Line# 302

Line 306: …first, Rural incomes ( remove capital ;R” in Rural and use small ‘r’. cross check other sentences for such cases in the entire work.

Response: Incorporated. Please see Line# 308. Whole manuscript is cross check and incorporated the changes where necessary.

Line 338: the results can have two compensations ( remove compensation and replace with ‘possible interpretations’)

Response: Incorporated. Please see Line# 340

Line 341; …. This might be the reason that there is well established ( I suggest ….’It could be because there is well established…).

Response: Incorporated. Please see Line# 343-344

Line 345: Secondly, waste management is a need rather than choice… ( I suggest: waste management service is important, so people spend money for it…)

Response: Incorporated. Please see Line# 347-348

Line 347-348 ( is Akhter’s finding in relation to a month or a year? It is good to state whether it is USD 4.8 per month or per year.)

Response: Akhtar’s study is based survey data in elite areas of Lahore. Although he did not mentioned either it was monthly or yearly but we perceived from his study it is monthly expenses on waste disposal services. See Line# 349-350

Line 370 -373 Please cross check the information in these sentences with what you have in the figure. In the discussion you have 83% with waste collection bin and 17% without waste collection bill, but your figure is saying the opposite. There is inconsistency. In figure 9, you have bin/ waste collection point not available/accessible 83% but the arrow is pointing to the small portion of the pie. You have bin/waste collection point is available /accessible by 13% but the arrow is pointing to larger portion of the pie.

Response: We revised the figure 9. Thank you for highlighting it. See line# 372-375 as well.

Line 394 -397: is it possible for you to find the mean/ average time for all respondents to get to the bin/collection point? That will help you if really most spend a lot of time disposing waste. From literature, is there a recommended time to be spent disposing waste? Beyond which time it is regarded as too much time? If there is such information and you can cite it, it will make your discussion more interesting.

Response: Dear Reviewer, thank you for your insightful comment. Unfortunately, we are unable to calculate the mean or average time for all respondents to reach the bin/collection point because the data was collected using a Likert scale, which does not provide precise numerical values for such calculations. However, we acknowledge the importance of this aspect and we discussed relevant literature to offer insights into the time spent disposing of waste. (For details see line 386-391)

Line 439-617: please carefully make sentence corrections to convey intended meanings to readers.

Response: We revised major portion of manuscript based on new variables. We highlighted this in green. Please let us know where we can further improve it.

Line 437: Factors affecting the households cash payments for waste collection and

disposal services

Response: Our heading is exactly like this. Please let us know what correction required in “Factors affecting the households cash payments for waste collection and

disposal services”

3. The regression reported here is only the robust least squares. Is it possible to report the OLS that was tried before testing for heteroskedasticity. It can be reported in same tables, side by side with the robust least squares. It will help readers to appreciate what you stated about the OLS and the robust least squares.

Response: Special thanks for this comment, as per your suggestion, we reported the results of OLS side by side with robust least square (RLS) results in the two tables (for details see Table 2 and Table 3).

4. Please check out your sentences especially in the discussion of the regression results.

Response: We revised major portion of manuscript based on new variables. We highlighted this in green. Please let us know where we can further improve it.

---

## [Editor Report · Decision Letter 2]

Households' expenditures for solid waste management services: influencing factors and deep insight

PONE-D-24-31637R2

Dear Dr. Ghulam,

We’re pleased to inform you that your manuscript has been judged scientifically suitable for publication and will be formally accepted for publication once it meets all outstanding technical requirements.

Kind regards,

Hatime Kamilcelebi

Academic Editor

PLOS ONE
---

## [Editor Report · Acceptance letter]

PONE-D-24-31637R2

PLOS ONE

Dear Dr. Mustafa,

I'm pleased to inform you that your manuscript has been deemed suitable for publication in PLOS ONE. Congratulations! Your manuscript is now being handed over to our production team.

Kind regards,

on behalf of

Dr. PLOS Manuscript Reassignment

Staff Editor

PLOS ONE